# A NAC-EXPANSIN module enhances maize kernel size by controlling nucellus elimination

Qin Sun [1], Yunfu Li [1], Dianming Gong[1], Aoqing Hu[1], Wanshun Zhong [1], Hailiang Zhao[1], Qiang Ning[1], Zengdong Tan[1], Kun Liang[1], Luyao Mu[1], David Jackson [1,2], Zuxin Zhang [1,3], Fang Yang [1,3] ✉ & Fazhan Qiu [1,3] ✉

Maize early endosperm development is initiated in coordination with elimination of maternal nucellar tissues. However, the underlying mechanisms are largely unknown. Here, we characterize a major quantitative trait locus for maize kernel size and weight that encodes an EXPANSIN gene, *ZmEXPB15*. The encoded β-expansin protein is expressed specifically in nucellus, and positively controls kernel size and weight by promoting nucellus elimination. We further show that two nucellus-enriched transcription factors (TFs), *ZmNAC11* and *ZmNAC29*, activate *ZmEXPB15* expression. Accordingly, these two TFs also promote kernel size and weight through nucellus elimination regulation, and genetic analyses support their interaction with *ZmEXPB15*. Importantly, hybrids derived from a *ZmEXPB15* overexpression line have increased kernel weight, demonstrates its potential value in breeding. Together, we reveal a pathway modulating the cellular processes of maternal nucellus elimination and early endosperm development, and an approach to improve kernel weight.

Maize (*Zea mays*) is one of the most important crops with increasing global demand for food and renewable energy. Kernel size and weight are key components for crop yield-component and are also important domestication and breeding traits[1]. To dissect the genetic basis of maize kernel-related traits, including kernel length, width and thickness, hundred kernel weight (HKW) and unit weight, hundreds of quantitative trait loci (QTLs) have been identified (www.maizegdb.org/qtl). However, only very few were cloned and characterized. The kernel size-related QTL, *qKW9*, was recently found to encode a penta-tricopeptide repeat (PPR) protein that affects photosynthesis and grain filling[2]. *ZmVPS29*, homologous to the *Arabidopsis Vacuolar Protein Sorting29* (*VPS29*), is a candidate kernel morphology QTL, *qKM4.08*. Overexpression of this gene produces slender kernels, but increases yield per plant[3]. Quite some kernel-related genes have been cloned from classical mutants with defects in embryo and/or

endosperm, such as defective kernel (*dek*), empty pericarp (*emp*), and small kernel (*smk*), and usually develop small and/or lethal kernels with limited potential for application[4–8]. Kernel traits are achieved through the coordinated growth of three genetically different components: embryo, endosperm, and the surrounding maternal tissues, including nucellus and pericarp[9]. The maternal nucellar tissue, originating from the ovule, acts as a bridge for apoplastic intercellular exchange between the mother plant and its developing offspring. It also provides an environment for the developing zygote and filial tissue[10,11]. However, the genetic and molecular regulation of maternal tissues and their impacts on maize kernel development are largely unknown.

Maize kernel development can be divided into three phases: early development, filling and maturation[10]. At an early stage, the maternal nucellar tissue degenerates by programmed cell death (PCD), and the contents of the dying cells are re-mobilized to feed the developing

[1]National Key Laboratory of Crop Genetic Improvement, Huazhong Agricultural University, 430070 Wuhan, Hubei, China. [2]Cold Spring Harbor Laboratory, Cold Spring Harbor, New York, NY 11724, USA. [3]Hubei Hongshan Laboratory, 430070 Wuhan, Hubei, China. ✉e-mail: fyang@mail.hzau.edu.cn; qiu-fazhan@mail.hzau.edu.cn

embryo and endosperm[12,13]. Interruption in nucellus PCD could lead to defective endosperm development. In Arabidopsis, the prolonged existence of nucellus due to defective degeneration in *transparent testa 16* (*tt16*) mutant seeds leads to a loss or misplaced chalazal endosperm cyst, revealing the importance of nucellus elimination and its feedback role in timing of endosperm development[11]. It has thus been proposed that endosperm and nucellus undergo antagonistic development at the early stage of Arabidopsis seed development. The rice *TT16* ortholog, *MADS29*, is expressed in the nucellus and regulates nucellus PCD by promoting the expression of PCD-related genes responsible for the cleavage of storage proteins[14–16]. The antagonistic development of nucellus and endosperm also occurs in rice, as suppression of *MADS29* expression in nucellus impairs endosperm growth and starch accumulation[15–18]. However, whether and how agonistic development process of nucellar tissue can be genetically manipulated to promote the endosperm growth and ultimately increase kernel weight remains unexplored.

In this work, we identify *ZmEXPB15* as a major QTL for kernel size and weight in maize, and find that it encodes an expansin protein that controls kernel size and weight through coordinating nucellus elimination and early endosperm development. The expression of *ZmEXPB15* is under the direct control of two nucellus-enriched genes, *ZmNAC11/ZmNAC29*, which control kernel size by a similar mechanism. Further, increased kernel weight of hybrids overexpressing *ZmEXPB15* demonstrates its potential application value of maize breeding.

## Results

### HKW9 acts maternally to control kernel size and weight

A major QTL for kernel size and weight, designated here as hundred kernel weight 9 (*HKW9*), was previously mapped to chromosome 9 (9.03-9.04 bin), using an $F_{2:3}$ population derived from two maize elite inbred lines with different kernel size, V671 and Mc[19]. To precisely estimate the genetic effect of *HKW9* on kernel traits, we generated near-isogenic lines (NILs), HKW9[Mc] and HKW9[V671], harboring the *HKW9* locus from Mc and V671, respectively (Fig. 1a). Over successive years of field trials, the large-kernel NIL, HKW9[Mc] showed a significant increase in HKW by 3.8%, in kernel length by 4.2%, in kernel width by 5.9% and in ear weight by 17.6%, compared to small-kernel NIL, HKW9[V671] (Fig. 1b–e). These results demonstrated that the *HKW9* QTL was responsible for multiple maize kernel related traits. Meanwhile, plant architecture, other ear traits and kernel starch accumulation were not significantly altered (Supplementary Fig. 1), suggesting a specific effect of *HKW9* on kernel size.

To ask whether the increased kernel size of HKW9[Mc] was due to a maternal effect, reciprocal crosses were performed. The HKW of $F_1$ kernels from the large-kernel NIL, HKW9[Mc], pollinated by HKW9[V671], was significantly larger than that of small-kernel NIL, HKW9[V671], pollinated by HKW9[Mc] (Fig. 1f). In addition, the HKW of HKW9[Mc]/HKW9[V671] and HKW9[V671]/HKW9[Mc] $F_2$ kernels was similar to that of HKW9[Mc]/HKW9[Mc] $F_2$ kernels and larger than that of HKW9[V671]/HKW9[V671] $F_2$ kernels (Fig. 1g). Together, these observations indicate that the *HKW9* QTL regulates kernel weight through a maternal effect.

### ZmEXPB15 is the candidate gene for HKW9

The *HKW9* QTL maps close to the centromere, which made fine mapping infeasible. We therefore applied transcriptome sequencing to assist in candidate gene isolation using 6 days after pollination (DAP) kernels, the stage when maternal tissues have a significant effect on kernel development. Within the 73-Mb (flanked by markers *umc2370* and *bnlg1209*) QTL mapping region[19], 27 of the annotated genes were differentially expressed between the two NILs (Supplementary Data 1), and two of them were highly and specifically expressed in developing seed based on publicly available RNA-seq data (Supplementary Fig. 2)[20]. These two genes, *Zm00001d045792* and *Zm00001d045861*, both encoding β-expansin, showed high (~98%) similarity in the coding

regions and were designated as *ZmEXPB14* and *ZmEXPB15* hereafter, respectively. By resequencing the coding regions, we found that *ZmEXPB14* contained only two nonsymponous mutations in the non-conserved sites, whereas *ZmEXPB15* covered a 4-bp deletion in the conserved domain of the last coding exon, leading to a frame shift of the last 60 amino acids and an addition of 24 residues in the C-terminus (Fig. 1h, Supplementary Fig. 3a, b). Thus, our main focus was first on *ZmEXPB15*. We further detected eight SNPs (at −740, −721, −562, −540, −482, −437, −286 and −283 bp) and one InDel (at −576 bp) within -1 kb promoter region upstream of the *ZmEXPB15* ATG. Consistently, the *ZmEXPB15* transcript was found to be differentially expressed between the two NILs from an early stage (-2 DAP) of kernel development, with a higher *ZmEXPB15* expression in the large-kernel NIL, HKW9[Mc] (Fig. 1i). Association analysis showed that *ZmEXPB15* expression was positively correlated with HKW (r = 0.45, $P$ < 0.0001, Fig. 1j) in 75 diverse inbred lines (Supplementary Data 2). To further investigate whether the promoter SNPs were responsible for the difference in HKW, we resequenced the -1 kb promoter region in 220 diverse inbred lines (Supplementary Data 2), and found that five SNPs (at −740, −540, −482, −286 and −283 bp) in the promoter regions significantly associated with HKW ($p$ = 3.73E-03, Fig. 1k, Supplementary Table 1). These five associated sites, which were in complete linkage disequilibrium, formed two haplotypes: one with HKW9[Mc] genotype and the other with HKW9[V671] genotype were named *Hap1* and *Hap2*, respectively (Fig. 1k). Inbred lines carrying *Hap1* had higher *ZmEXPB15* expression and larger kernels than those carrying *Hap2* (Fig. 1l, Supplementary Fig. 4), consistent with the positive correlation of the *ZmEXPB15* expression with the HKW.

To further verify the candidate gene for the QTL *HKW9*, we generated knockout mutants for both *ZmEXPB14* and *ZmEXPB15* genes using a CRISPR/Cas9 strategy (Supplementary Fig. 5a). Only double mutants (KO1, KO2, KO3) were obtained with two genes being edited together due to their high similarity (Supplementary Fig. 5b). We next separated the *zmexpb14* and *zmexpb15* mutations by screening a large (>30,000) segregating population due to their close linkage across the centromere. Single *zmexpb14* mutant did not show an obvious change in HKW, whereas *zmexpb15* decreased the HKW significantly (Fig. 1m, Supplementary Fig. 5c). Importantly, the double *zmexpb14;zmexpb15* mutant did not show any enhancement in the HKW difference, compared to the *zmexpb15* single mutant in which *ZmEXPB14* expression was not altered (Fig. 1m, Supplementary Fig. 5d), indicating that *ZmEXPB15* is the major player for the HKW difference. Taken together, our findings suggest that *ZmEXPB15* (*Zm00001d045861*) is the candidate gene for the *HKW9* QTL, and positively control kernel weight.

### Functional validation of ZmEXPB15

To verify the biological function of *ZmEXPB15* in control of kernel size and weight, we investigated three knockout alleles (KO1, KO2, KO3), in which different truncations nearby the ATG led to early frame shifts (Supplementary Fig. 5b). All three alleles had lower HKW and smaller kernel size (Fig. 2a–i) than their corresponding wild-type (WT) plants. To further validate the effect of *ZmEXPB15*, we overexpressed the *ZmEXPB15* large-kernel allele coding sequence using a maize ubiquitin promoter. Three independent overexpression lines (OE1, OE2, OE3) with significantly increased expression level of *ZmEXPB15* (Supplementary Fig. 5e) had an increase in HKW and larger kernel length and width, compared to the corresponding non-transgenic lines (Fig. 2j–r). All these results consistently demonstrate that *ZmEXPB15* positively controls kernel size and weight in maize.

Next, we evaluated the potential application of *ZmEXPB15* in molecular breeding, by crossing the OE1 overexpression line (as female parent) by four different elite inbred lines. All four $F_1$ hybrids harboring transgenic positive plants had significantly higher HKW, compared to control non-transgenic hybrids (Fig. 2s, t). The increase in HKW ranged from 2.8% to 12.4%, with 6.3% on average. The kernel length and width

also increased, although the different hybrids varied in which size trait was altered (Supplementary Fig. 6a, b). In addition, most hybrids did not affect other traits, such as ear length (Supplementary Fig. 6c–e). Consistent with our finding that *ZmEXPB15* acts maternally to regulate

kernel weight, most reciprocal cross hybrids using OE1 as the male parent did not show a significant change in HKW (Supplementary Fig. 6f). Thus, *ZmEXPB15* showed potential application in enhancing kernel weight.

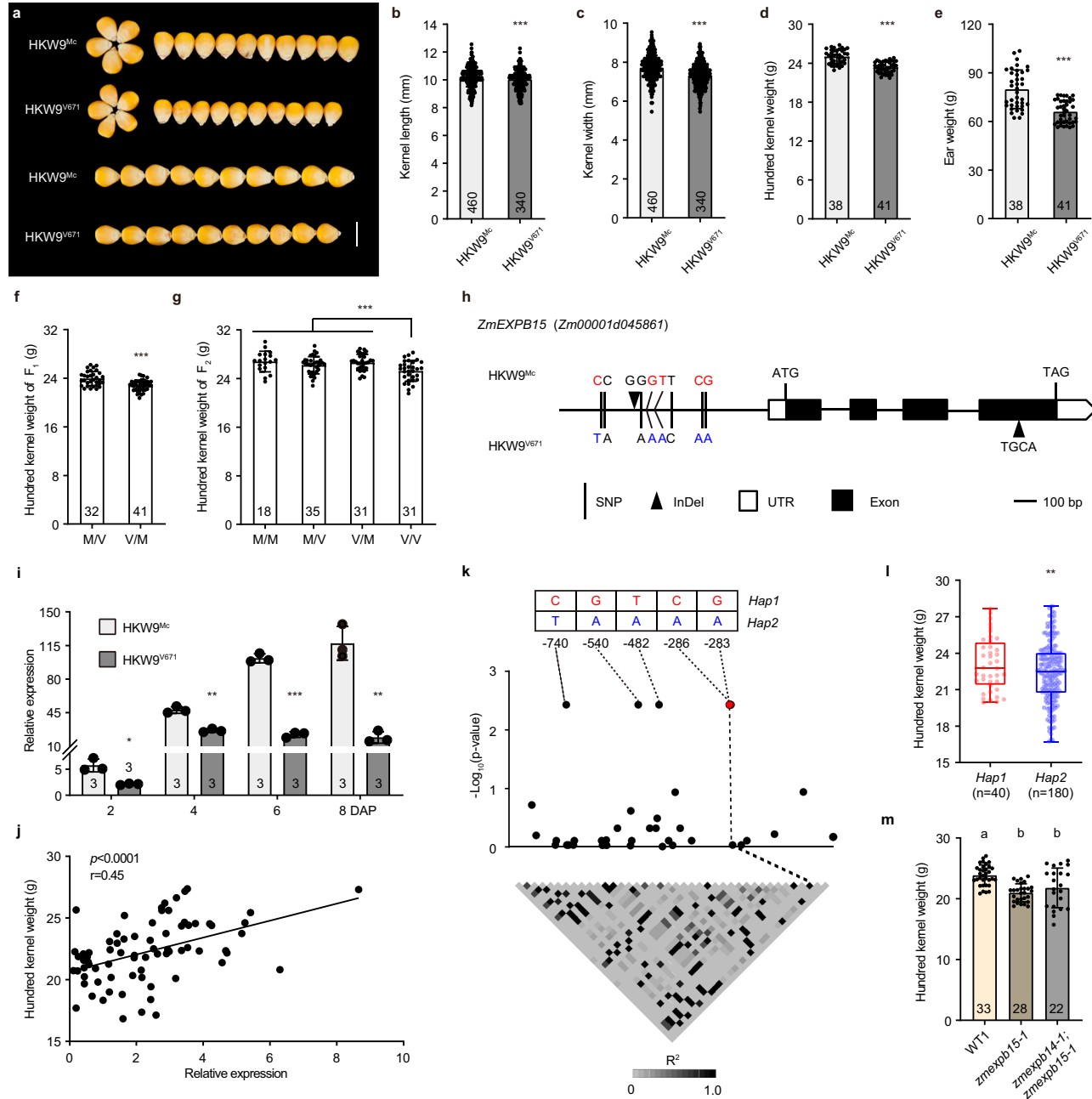

**Fig. 1 | *HKW9* acts maternally to control kernel size and weight. a** Comparison of mature kernels between HKW9$^{Mc}$ and HKW9$^{V671}$. Scale bar = 1 cm. **b–e** Quantitative analysis of kernel length (**b**), kernel width (**c**), hundred kernel weight (**d**) and ear weight (**e**) between HKW9$^{Mc}$ and HKW9$^{V671}$. **f, g** Quantitative analysis of hundred kernel weight (HKW) of F$_1$ kernels (**f**) from the HKW9$^{Mc}$/HKW9$^{V671}$ (M/V) and HKW9$^{V671}$/HKW9$^{Mc}$ (V/M) plants, and F$_2$ kernels (**g**) from the HKW9$^{Mc}$/HKW9$^{Mc}$ (M/M), M/V, V/M and HKW9$^{V671}$/HKW9$^{V671}$ (V/V) plants. **h** Gene structure of the candidate gene, *Zm00001d045861* (*ZmEXPB15*), and the polymorphisms between HKW9$^{Mc}$ and HKW9$^{V671}$. Eight SNPs (black lines) and one 1-bp InDel (black triangle) in the promoter and one 4-bp InDel (black triangle) in the last exon are shown; white boxes, untranslated regions; black boxes, coding sequences. **i** Expression of *ZmEXPB15* is higher in large kernel NIL HKW9$^{Mc}$ in early developing kernels. All expression levels from three biological repeats were normalized to *Actin*. **j** Pearson correlation between *ZmEXPB15* expression and the HKW among 75 inbred lines. **k** The

association analysis of *ZmEXPB15* promoter variations (−843 to −24 relative to ATG) with HKW. The white to black heat-map shows the pairwise linkage disequilibrium pattern by R$^2$; the genotypes and positions of five significant association loci are shown on the top. **l** Box-and-whisker plots of HKW for the *Hap1* (red box, *n* = 40) and *Hap2* (blue box, *n* = 180). Each box represents the median and interquartile range, and whiskers extend to maximum and minimum values. **m** Quantitative analysis of the HKW in *zmexpb15-1* single and *zmexpb14-1;zmexpb15-1* double mutants compare to the corresponding wild-type (WT1). Number on each column is the sample size. Values in (**b–g, i, l, m**) are means ± s.d. (standard deviation), and the significance in (**b–g, k–m**) is estimated by one-way ANOVA, and the significance in (**i, j**) is estimated by a two-tailed Student's *t* test. *$P$ < 0.05, **$P$ < 0.01, ***$P$ < 0.001. The Tukey HSD test is used in (**m**) and statistical differences ($P$ < 0.05) are indicated by different letters. Source data are provided as a Source Data file.

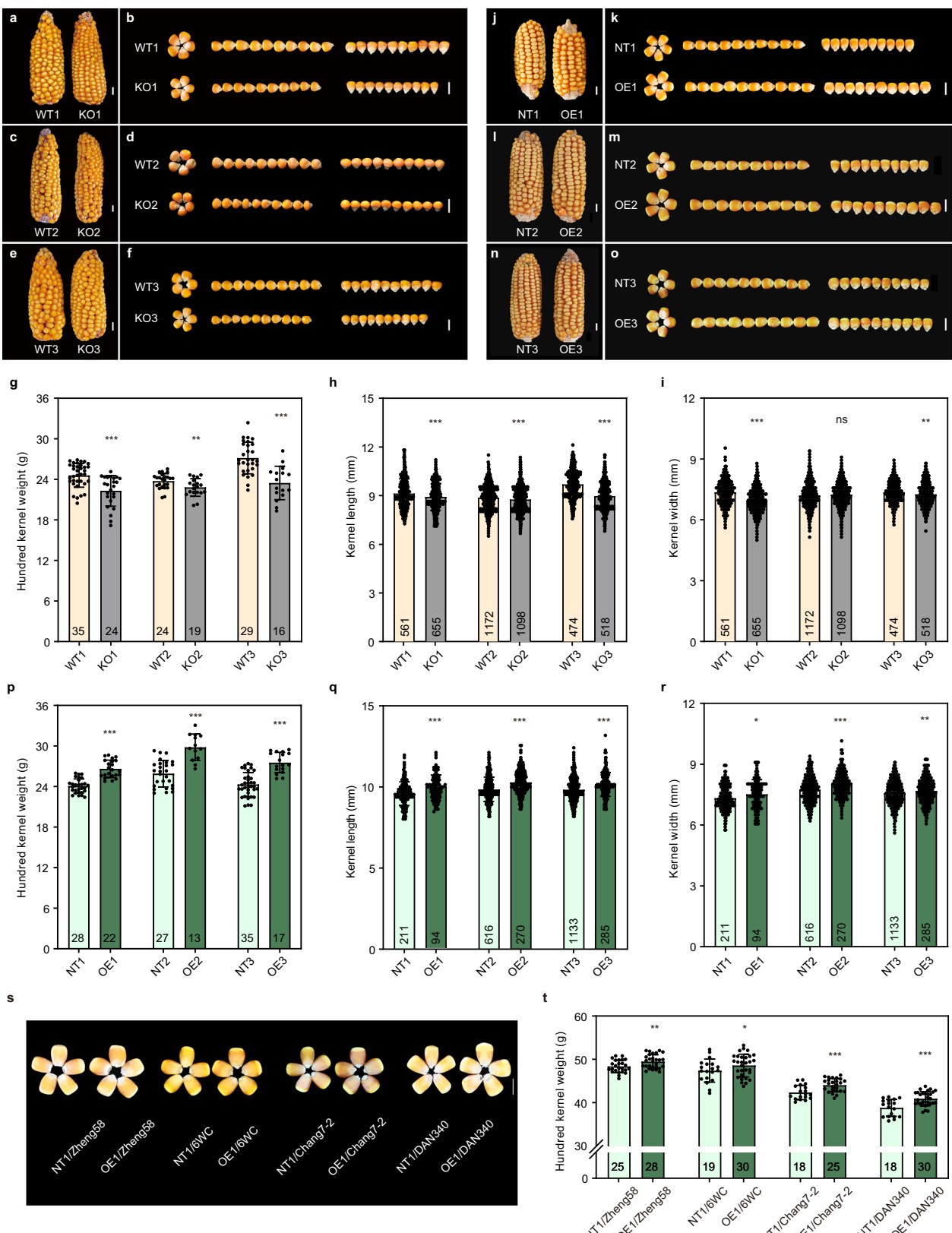

### ZmEXPB15 is specifically expressed in nucellus of developing kernel

The maternal role of *ZmEXPB15* in the control of kernel traits indicated that it might be expressed and function in maternal tissues during early kernel development, and was supported by its high expression in the nucellus in publicly available RNA-seq data (Supplementary Fig. 7)[20]. This expression profile was confirmed by our real-time quantitative reverse transcription PCR (qRT-PCR) analyses, showing that *ZmEXPB15* was exclusively expressed during the early developmental stages of 2–8 DAP, with the highest level at 4 DAP, when nucellar tissue is enriched (Fig. 3a). To investigate the expression pattern more carefully, RNA in situ hybridization using an antisense

**Fig. 2 | *ZmEXPB15* positively control kernel size and weight. a–f** Three knockout lines (KO1, KO2 and KO3) with loss-of-function mutations in *ZmEXPB15* had smaller kernels than the corresponding wild-type (WT1, WT2 and WT3) in a C01 inbred background. **g–i** Quantitative analysis of the hundred kernel weight (**g**), kernel length (**h**) and kernel width (**i**) in three knockout lines. **j–o** Three *ZmEXPB15* over-expressing lines (OE1, OE2 and OE3) had larger kernels than their corresponding non-transgenic lines (NT1, NT2 and NT3) in a B104 inbred background. **p–r** Quantitative analysis of the hundred kernel weight (**p**), kernel length (**q**) and kernel width (**r**) in three overexpressing lines. **s** Mature kernels of four F$_1$ hybrids derived from overexpression line OE1 and the corresponding control NT1 by crossing to four elite inbred lines (Zheng58, 6WC, Chang7-2, Dan340). **t** Quantitative analysis of the HKW of four F$_1$ hybrids same as (**s**). Number on the bottom of each column is the sample size. Values in (**g–i**, **p–r**, **t**) are means ± s.d., and the significance is estimated by the one-way ANOVA. *$P < 0.05$, **$P < 0.01$, ***$P < 0.001$. ns, non-significance. Scale bar = 1 cm in (**a–f**, **j–o**, **s**). Source data are provided as a Source Data file.

probe against *ZmEXPB15* was performed at 2 to 12 DAP stages, when the maternal nucellar tissue was still present. A specific expression pattern in the nucellus was observed, starting early from 2 DAP and gradually decreasing after 8 DAP, when the endosperm started to rapidly expand (Fig. 3b). We also generated a transgenic line harboring ZmEXPB15 fused with the green fluorescent protein (GFP) under its native promoter to detect its subcellular localization. A strong GFP signal was observed in the nucellus (Fig. 3c) and the signal gradually decreased, with faint florescence at 8 DAP, consistent with its dynamic transcript distribution shown by in situ hybridization. After plasmolysis, the GFP signal was clearly detected in the cell wall, cytoplasm and also nucleus (Fig. 3d). Our results demonstrate that *ZmEXPB15* was specifically expressed in nucellus, and explain its maternal effect on kernel development.

### *ZmEXPB15* promotes nucellus cell expansion and elimination

To investigate how *ZmEXPB15* affects the nucellus development, we compared the nucellus size of the two NILs, and found that the nucellus in the large-kernel line HKW9$^{Mc}$ with higher *ZmEXPB15* expression was thicker than that in HKW9$^{V671}$ (Fig. 4a). The cell area, cell length and width of nucellar cells in HKW9$^{Mc}$ were significantly larger than those in HKW9$^{V671}$, while the cell number was compatible (Fig. 4b–g; Supplementary Fig. 8a). These observations indicated that *ZmEXPB15* could promote nucellus cell expansion, leading to a larger nucellus volume. During early kernel development, nucellar tissue undergoes a degenerative process recognized as programmed cell death (PCD), accompanied by the expansion of endosperm tissues[21]. These two antagonistic processes occur in a coordinated manner[22]. To investigate how *ZmEXPB15* coordinates the nucellus and endosperm development, the areas of both tissues were measured in the two NILs from 0 to 12 DAP. There was no difference at 0 DAP with 100% area of nucellus, however, the ratio of nucellus to endosperm decreased differentially between the two NILs starting from 3 DAP, with a significantly lower ratio in the large-kernel HKW$^{Mc}$ line than that in the small-kernel HKW$^{V671}$ line (Fig. 4h, i). At 12 DAP, the ratio of nucellus to endosperm area in HKW9$^{Mc}$ was around 25%, in contrast to around 50% in HKW$^{V671}$, suggesting a faster nucellus elimination and endosperm expansion present in HKW9$^{Mc}$ that showed a higher *ZmEXPB15* expression level (Fig. 4i). Consistently, the same effect of *ZmEXPB15* on the coordinated nucellus and endosperm development process was observed in the *ZmEXPB15* knockout (KO1) line with a higher ratio of nucellus to endosperm area than the control, and also in the overexpression (OE1) line which showed the opposite (Supplementary Fig. 8b). These results suggested that *ZmEXPB15* was involved in the nucellus elimination which coordinates endosperm development.

To confirm the speculation above, we performed a terminal deoxynucleotidyl transferase dUTP nick-end labeling (TUNEL) assay to monitor how the nucellus elimination process was regulated by *ZmEXPB15*. In the large-kernel line HKW$^{Mc}$ in which *ZmEXPB15* was highly expressed, the TUNEL-positive nuclei were detected in the nucellar cells of as early as 2 DAP, and increased at later stages. In contrast, the TUNEL signals in the small-kernel line HKW$^{V671}$were almost undetectable at 3 DAP (Fig. 4j), although the distribution of TUNEL-positive nuclei became undistinguishable at 6 DAP between the two NILs (Fig. 4j). Further, the *ZmEXPB15* knockout (KO1) line showed

an obviously delayed appearance of the TUNEL-positive nuclei, and the *ZmEXPB15* overexpression (OE1) line showed the opposite, compared to the controls (Supplementary Fig. 8c, d). These results indicated that *ZmEXPB15* could promote nucellus PCD process. In addition, Evan's blue staining, as another more representative marker for cell death, showed that the nucellar cells of large-kernel line HKW$^{Mc}$ had significantly deeper staining than those in small-kernel line HKW$^{V671}$ at the stage of 4 and 6 DAP (Supplementary Fig. 8e). Consistently, the *ZmEXPB15*-overexpressed line (OE1) also stained deeper than the control as early as 2 DAP (Supplementary Fig. 8f). These observations revealed that *ZmEXPB15* promoted the PCD process and cell death of nucellus to ensure a timely endosperm development, which was likely responsible for the increased kernel size and weight.

Seed developmental PCD involves diverse classes of proteases, including cysteine proteases, serine proteases, and aspartic proteases[23]. We next compared the expression of some annotated protease genes between the two NILs, including vacuolar processing enzymes 4 (*VPE4*) homologous to barley *VPE4*, which has been known to execute programmed cell death in barley pericarp[24,25], cysteine protease 5 (*CCP5*) homologous to *Arabidopsis* xylem cysteine proteinase 1 (*XCP1*), which promote catabolism in tracheary elements during xylogenesis[26], serine proteases1 (*SER1*) homologous to *Arabidopsis* senescence-associated gene *SAG15*, which functions in protein degradation in senescing chloroplasts[27], aspartyl protease1 (*AED1*) homologous to *Arabidopsis* aspartyl protease 1 (*AED1*), which may degrade apoplastic proteins[28]. All these potentially PCD-related genes were expressed at significantly lower levels in 4-DAP kernels of HKW$^{V671}$ than in HKW$^{Mc}$ (Fig. 4k). This result is in line with the delayed PCD in HKW$^{V671}$ in which *ZmEXPB15* was expressed at a lower level. Taken all together, we propose that *ZmEXPB15* positively controls early kernel development by promoting the nucellus PCD and coordinating the antagonistic development between nucellus and endosperm.

### ZmNAC11 and ZmNAC29 may activate the expression of *ZmEXPB15*

Our association analysis and the correlation of *ZmEXPB15* expression level with kernel weight prompted us to speculate that specific *cis*-elements might regulate the differences in *ZmEXPB15* expression between the two NILs. We thus investigated the promoter region of *ZmEXPB15* and found that several CACG motifs, known binding sites for NAC transcription factors (TFs)[29,30], were enriched (Fig. 5a). Two SNPs at positions of −286 bp and −283 bp upstream of the ATG resided in one CACG motif, which was also associated with HKW (Fig. 1k). To search for the potential NAC TFs binding to the promoter of *ZmEXPB15*, we looked at nucellus transcriptome data from 8-DAP maize kernels[31], and found that *ZmNAC11* and *ZmNAC29*, two close paralog genes, were highly expressed in nucellus (Supplementary Fig. 9). Our qRT-PCR and mRNA in situ analysis confirmed that these two NAC TFs were predominantly expressed in the nucellus and reached a peak in 4-DAP kernels, a similar expression pattern to *ZmEXPB15* (Fig. 5b, c).

To investigate the DNA binding of *ZmNAC11* and *ZmNAC29* to the promoter of *ZmEXPB15*, electrophoresis mobility shift assays (EMSAs) were performed using a probe containing the CACG motif from the promoter of *ZmEXPB15*, which harbors the two

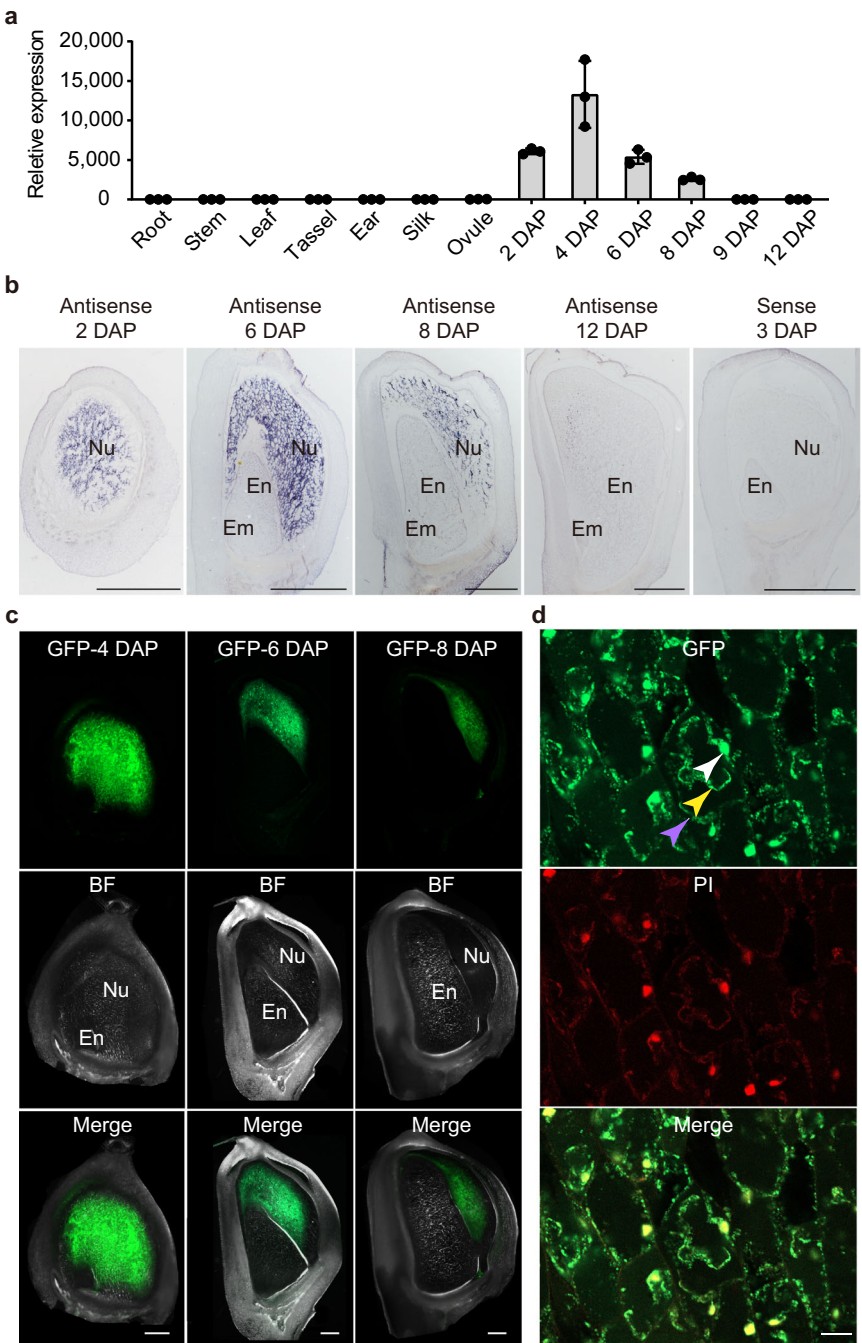

**Fig. 3 | *ZmEXPB15* accumulates specifically in nucellus of developing kernels.**
**a** qRT-PCR analysis of *ZmEXPB15* in root, stem, leaf, tassel, ear, silk and developing
kernels at 0 (ovule) to 12 DAP. All expression levels from three biological repeats
were normalized to *Actin*. Values are means ± s.d. **b** In situ localization of *ZmEXPB15*
mRNA in developing kernels at 2 to 12 DAP. Positive signals (shown in purple) are
clearly restricted to the nucellus at 2 to 8 DAP (first three panels) using the
*ZmEXPB15* antisense probe. No signal is observed in the section of a 12-DAP kernel
(fourth panel). The control was performed using 3-DAP kernel with a *ZmEXPB15*
sense probe (last panel). **c** Localization of ZmEXPB15-GFP in developing kernels at

4, 6 and 8 DAP from *proZmEXPB15:ZmEXPB15*-GFP transgenic plants. The
ZmEXPB15-GFP signal was specifically detected in the nucellus. BF bright-field
image, Merge merge of GFP and BF images. Nu nucellus, En endosperm, Em
embryo. **d** ZmEXPB15-GFP expression was observed in the cell wall (purple arrow),
cytoplasm (yellow arrow) and nucleus (white arrow) of the nucellar cells in 4-DAP
kernels. PI, propidium iodide staining. Merge, merge of GFP and PI images. The
experiments in (**b**–**d**) were repeated two times with a similar result. Scale bar = 1
mm in (**b**); 500 μm in (**c**); 20 μm in (**d**). Source data are provided as a Source
Data file.

SNPs present in two NILs (Fig. 5a). Strong shifts of the probe were
detected in the presence of the ZmNAC11 and ZmNAC29 recom-
binant proteins (Fig. 5d). Notably, the probe from the large-kernel
NIL, HKW^Mc, which contains the intact CACG motif, showed a
stronger shift than that from HKW^V671 containing two SNPs that
disrupted the motif (Fig. 5d). Next, we investigated the effects of
ZmNAC11 and ZmNAC29 proteins on the transcription activity of

*ZmEXPB15* using the luciferase (*LUC*) reporter in maize leaf pro-
toplasts. A 436-bp *ZmEXPB15* promoter containing two CACG
motifs was used to drive LUC (Supplementary Fig. 10a), and both
ZmNAC11 and ZmNAC29 activated *ZmEXPB15* expression (Fig. 5e).
Consistently, the promoter from HKW^Mc showed a significantly
stronger transcriptional activity in the presence of the NAC pro-
teins than that from HKW^V671 (Fig. 5e). Further, when a longer

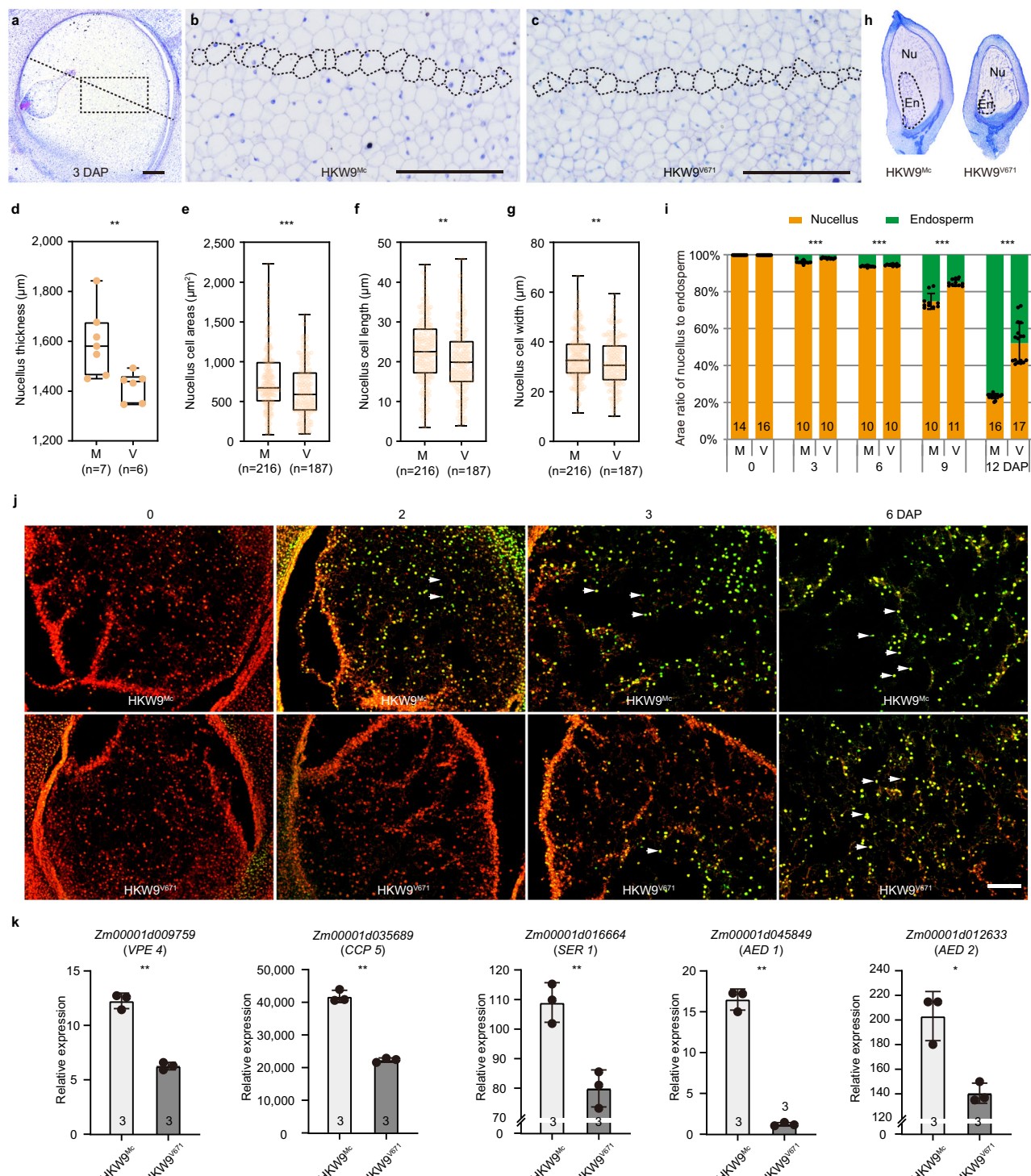

**Fig. 4 | Nucellus cell expansion and elimination is delayed in the small-kernel HKW9^V671 line. a** Longitudinal semi-thin section of 3-DAP kernels used for nucellus size measurement. The black dotted lines indicate the region where nucellus thickness was measured. **b, c** Enlarged image of the black boxed region in (**a**) of HKW9^Mc (**b**) and HKW9^V671 (**c**). A row of nucellar cells are framed by black-dots in (**b, c**) to measure the cell size. **d–g** Quantification of nucellus thickness (**d**), cell areas (**e**), cell length (**f**) and cell width (**g**) of HKW9^Mc (M) and HKW9^V671 (V). The samples in M and V are 7 and 6 in (**d**), and 216 and 187 in (**e–g**), respectively. The sample numbers in (**e–g**) refer to all cells counted along the black dotted lines from all independent samples. Each box represents the median and interquartile range, and whiskers extend to maximum and minimum values. **h** Toluidine blue stained sections of NILs kernels at 9 DAP. Nu, nucellus; En, endosperm. **i** Ratio of nucellus (orange box) to endosperm area (green box) in developing kernels of M and V.

**j** TUNEL assays of the nucellus in the developing kernels from NILs. Nuclei were prestained with propidium iodide (PI) seen as red fluorescence. The green and yellow fluorescence correspond to the TUNEL-positive nuclei (partly marked white arrows). **k** qRT-PCR analysis of PCD-related genes in the NILs kernels at 4 DAP. *Zm00001d009759* for vacuolar processing enzymes 4 (*VPE4*); *Zm00001d035689* for cysteine protease 5 (*CCP5*); *Zm00001d016664* for serine proteases 1 (*SER1*); *Zm00001d045849* for aspartyl protease 1 (*AED1*) and *Zm00001d012633* for aspartyl protease 2 (*AED2*). All expression levels from three biological repeats were normalized to *Actin*. Number on the bottom of each column is the sample size. Values are means ± s.d., two-tailed Student's *t* test. *$P < 0.05$, **$P < 0.01$, ***$P < 0.001$. Scale bar = 200 μm in (**a–c**); 1 mm in (**h**); 500 μm in (**j**). Source data are provided as a Source Data file.

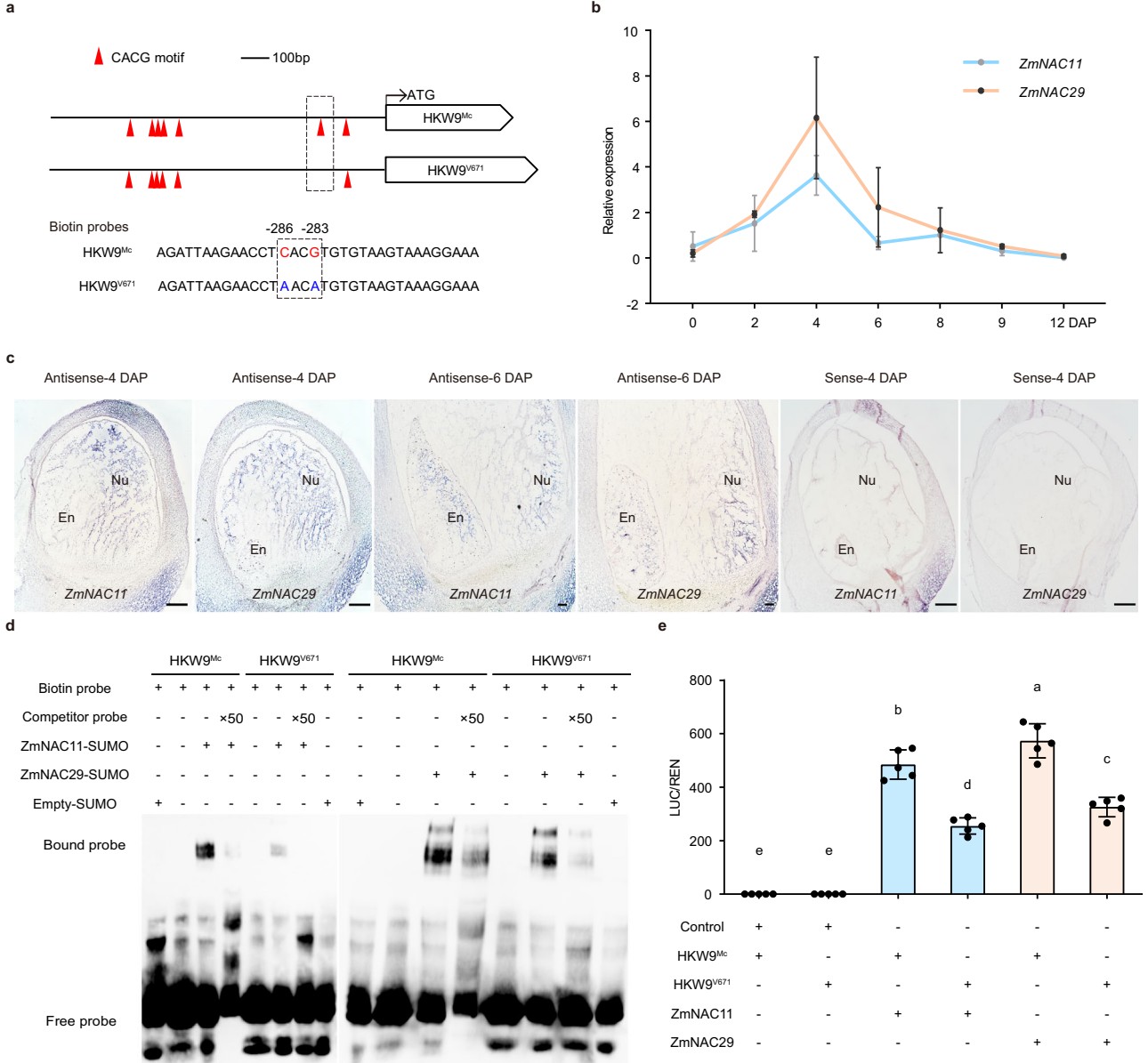

**Fig. 5 | Nucellus-expressed ZmNAC11 and ZmNAC29 bind to and activate ZmEXPB15. a** Schematic diagrams of the promoters of HKW9^Mc and HKW9^V671. The red triangles represent in vitro CACG motifs. The dashed box represents the positions of the biotin probes (**b**), qRT-PCR analysis of *ZmNAC11* and *ZmNAC29* expression during early kernel development of 0 to 12 DAP. All expression levels from three biological repeats were normalized to *Actin*. **c** In situ localization of *ZmNAC11* and *ZmNAC29* expression in the developing kenrels at 4 and 6 DAP. Positive signals were found in the nucellus (Nu) and endosperm (En) using the *ZmNAC11* and *ZmNAC29* antisense probes. The control was performed using 4-DAP kernel with *ZmNAC11* and *ZmNAC29* sense probes. Scale bars = 50 μm. **d** DNA binding affinities of the recombinant ZmNAC11 and ZmNAC29 proteins on the

CACG motif-containing promoter regions of HKW9^Mc and HKW9^V671 detected by electrophoresis mobility shift assays (EMSAs). ZmNAC11 and ZmNAC29 bind more strongly to the promoter fragment (−299 to −249 upstream of the ATG) from the large-kernel line HKW9^Mc than to that from the small-kernel line HKW9^V671. The unlabeled intact probes were used for competition. The experiment was repeated two times with a similar result. **e** Transactivation activities of the ZmNAC11 and ZmNAC29 proteins on *ZmEXPB15* promoters of two NILs (HKW9^Mc and HKW9^V671). Values are means ± s.d. (*n* = 5 biologically independent samples), Tukey HSD test is used and the statistical differences (*P* < 0.05, two-sided) are indicated by different letters. Source data are provided as a Source Data file.

1308-bp *ZmEXPB15* promoter, which contains another four CACG motifs, was used, *ZmEXPB15* transcription was significantly higher, suggesting a stronger binding of the two NAC proteins on the promoter (Supplementary Fig. 10b–d). When ZmNAC11 and ZmNAC29 proteins were coexpressed, the activation of *ZmEXPB15* was not obviously changed (Supplementary Fig. 10e), suggesting there was no interaction between these two NAC proteins. Our findings indicate that the nucellus-enriched ZmNAC11 and ZmNAC29 TFs could bind the promoter and activate the transcription of *ZmEXPB15*.

## Mutations in *ZmNAC11* and *ZmNAC29* cause similar kernel defects to *zmexpb15*

To investigate the significance of *ZmEXPB15* activation by ZmNAC11 and ZmNAC29, we investigated the roles of two NAC genes on kernel traits. Knockout lines were created using CRISPR-Cas9 in the maize C01 inbred line (Supplementary Fig. 11a). Two *zmnac11* knockout mutants (*zmnac11-1* and *zmnac11-2*) and three *zmnac29* knockout mutants (*zmnac29-1*, *zmnac29-2* and *zmnac29-3*) all resulted in deletion or frame-shift of amino acids in the conserved NAC domain (Supplementary Fig. 11b, c). Interestingly, each single mutant produced smaller

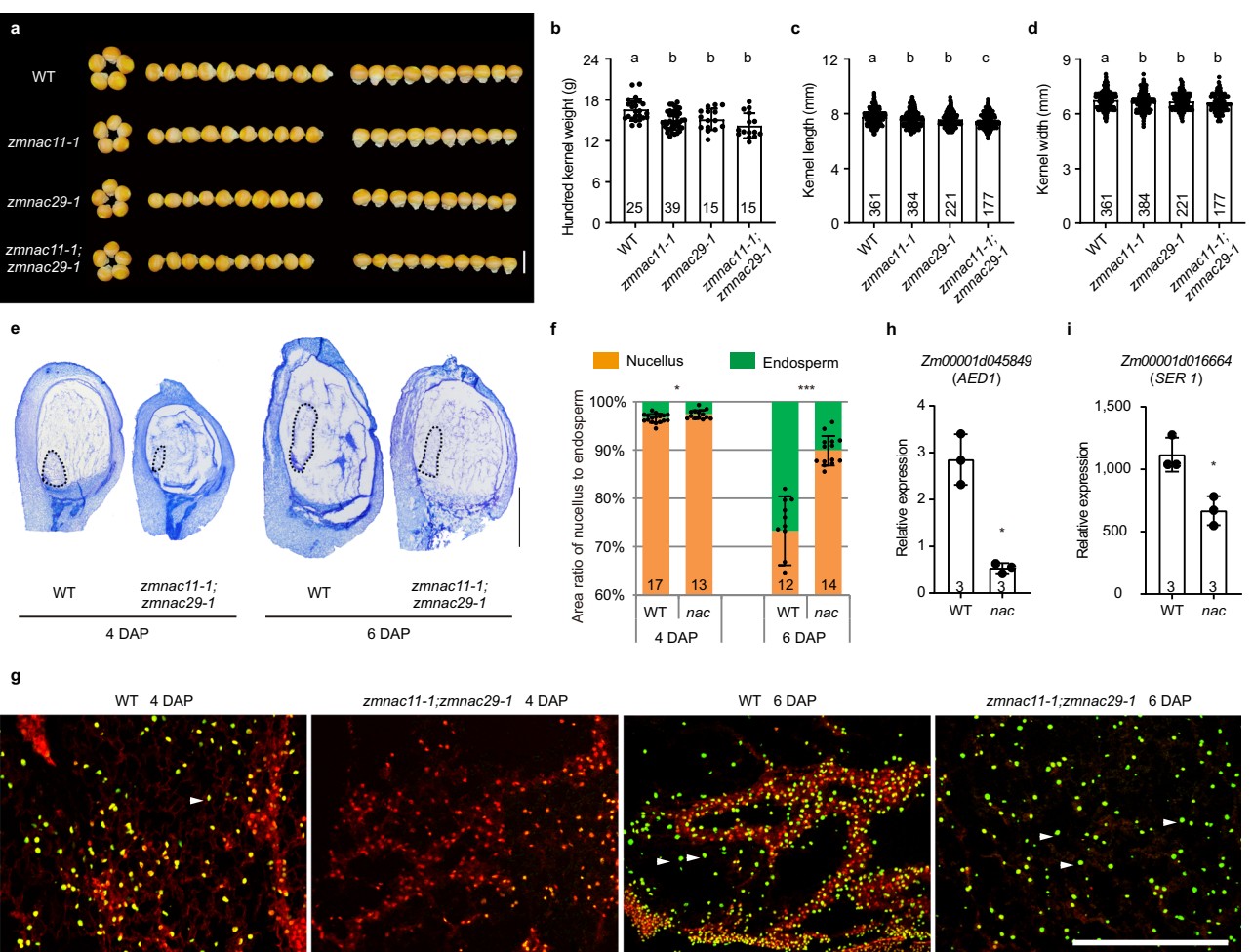

**Fig. 6 | Mutations in *ZmNAC11* and *ZmNAC29* cause similar defects to *zmexpb15* in kernel development. a** The mature kernels of *zmnac11-1*, *zmnac29-1* and *zmnac11-1;zmnac29-1* in comparison to the wild type (WT) in C01 background. **b–d** Quantitative analysis of hundred kernel weight (**b**), kernel length (**c**) and kernel width (**d**) of wild type (WT), *zmnac11-1*, *zmnac29-1* and *zmnac11-1;zmnac29-1*. **e** Toluidine blue sections of WT and *zmnac11-1;zmnac29-1* kernels at 4 and 6 DAP. **f** Ratio of nucellus (orange box) to endosperm area (green box) of 4-DAP and 6-DAP kernels in *zmnac11-1;zmnac29-1* (*nac*) double mutant in comparison to the wild type (WT) control. Longitudinal median sections with maximum area of the entire endosperm were examined and quantified. **g** TUNEL assays of the nucellus at 4 and 6 DAP of WT and *zmnac11-1;zmnac29-1* double mutant. WT kernel at 4 DAP has strong TUNEL-positive signals in nucellar cells (white arrow), whereas *zmnac11-1;zmnac29-1* kernel at 4-DAP does not show obvious TUNEL signal. The TUNEL-positive signals (partly marked white arrows) are indistinguishable at 6-DAP between *zmnac11-1;zmnac29-1* and WT. **h, i** RT-PCR analysis shows the down-regulation of two PCD-related genes in the *zmnac11-1;zmnac29-1* (*nac*) kernels at 4 DAP. All expression levels from three biological repeats were normalized to *Actin*. Scale bar = 1 cm in (**a**); 1 mm in (**e**); 500 μm in (**g**). Number on the bottom of each column is the sample size. Values in (**b–d**) are means ± s.d. and Tukey HSD test is used and statistical differences (*P* < 0.05, one-sided) are indicated by different letters. Values in (**f, h, i**) are means ± s.d., and the significance is estimated by a two-tailed Student's *t* test. *P* < 0.05, **P* < 0.01, ***P* < 0.001. Source data are provided as a Source Data file.

kernels with significantly decreased HKW, kernel length and width compared to the segregated wild-type sibs (Fig. 6a–d, Supplementary Fig. 11d–k). The *zmnac11-1;zmnac29-1* double mutant did not show a significant enhancement compared to each single mutant (Fig. 6a–d), suggesting that these two NACs may work in one pathway. This speculation was in line with the findings that *ZmNAC11* transcript was dramatically decreased in *zmnac29* mutant, whereas *ZmNAC29* was not affected by the loss of *ZmNAC11* mutation (Supplementary Fig. 12a, b). All these results demonstrate that *ZmNAC11* and *ZmNAC29* both positively control kernel size and weight, with a possibility that *ZmNAC29* may also work upstream of *ZmNAC11*.

To explore whether the kernel defects of *zmnac11* and *zmnac29* were also associated with nucellus, cytological sectioning and TUNEL assay were performed using the developing kernels of *zmnac11-1;zmnac29-1* double mutants. The mutants had significantly bigger nucellus and lower endosperm area at 4 and 6

DAP compared to WT sibs (Fig. 6e, f), indicating that nucellus elimination was delayed. In support of this, TUNEL-positive nuclei were barely detected in nucellar cells of 4-DAP *zmnac11-1;zmnac29-1* kernels, whereas the signal was strong in wild type nucellus (Fig. 6g). However, at a later stage, the TUNEL signals in 6-DAP nucellus was indistinguishable between *zmnac11-1;zmnac29-1* (*nac*) double mutant and wild type. Consistently, two PCD-related genes, *AED1* and *SER1*, were also significantly down-regulated in the *nac* kernels (Fig. 6h, i). In addition, the HKW of $F_1$ kernels from WT ear pollinated with the *nac* double mutant was significantly larger than that of *nac* ear pollinated with WT pollen (Supplementary Fig. 13a, b). The reciprocal cross results demonstrate that *ZmNAC11* and *ZmNAC29* control kernel development also maternally. Taken together, these observations suggest that *ZmNAC11* and *ZmNAC29* positively regulate kernel size and weight by promoting nucellus elimination, similar to *ZmEXPB15*.

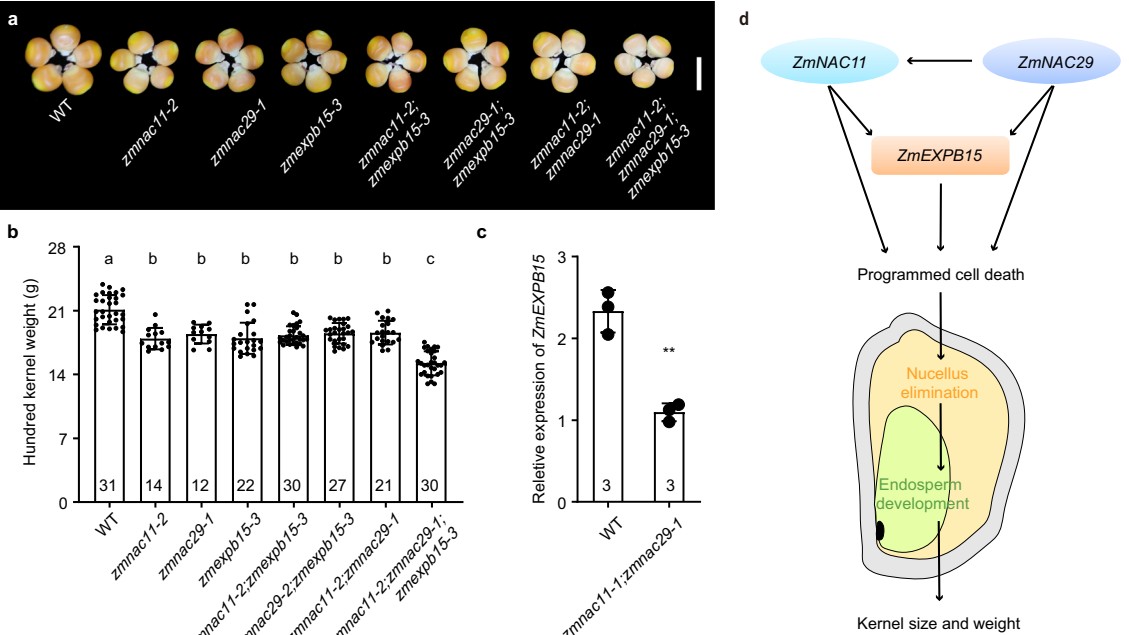

**Fig. 7 | *ZmEXPB15* may act in a common pathway with *ZmNAC11* and *ZmNAC29* to regulate kernel size and weight. a** Comparison of the mature kernels of wild type (WT) and different mutation combinations, *zmnac11-2*, *zmnac29-1*, *zmexpb15-3* (KO3), *zmnac11-2;zmexpb15-3*, *zmnac29-1;zmexpb15-3*, *zmnac11-2;zmnac29-1* and *zmnac11-2;zmnac29-1;zmexpb15-3* in C01 background. Scale bar = 1 cm.
**b** Quantitative analysis of HKW of WT, *zmnac11-2*, *zmnac29-1*, *zmexpb15-3*, *zmnac11-2;zmexpb15-3*, *zmnac29-1;zmexpb15-3*, *zmnac11-2;zmnac29-1* and *zmnac11-2;zmnac29-1;zmexpb15-3*. Value is means ± s.d. Tukey HSD test is used and statistical differences (*P* < 0.05, one-sided) are indicated by different letters. Number on the bottom of each column is the sample size. **c** qRT-PCR analysis of *ZmEXPB15* in WT,

*zmnac11-1* and *zmnac29-1* kernels at 4 DAP. All expression levels from three biological repeats were normalized to *Actin*. Value is means ± s.d., and the significance is estimated by a two-tailed Student's *t* test. ****P* < 0.001. **d** A proposed model for the NAC-ZmEXPB15 module regulating the maternal nucellar tissue to ensure kernel size and weight in maize. ZmNAC11 and ZmNAC29 could bind and activate *ZmEXPB15* to promote nucellus PCD process and further elimination, leading to an expansion and a normal growth of endosperm. Two NAC genes and ZmEXPB15 could also work independently in this process, and ZmNAC29 might work upstream of ZmNAC11. Source data are provided as a Source Data file.

## *ZmEXPB15* acts genetically with *ZmNAC11* and *ZmNAC29* to control kernel weight

To explore the genetic interaction between *zmexpb15*, *zmnac11* and *zmnac29*, the *zmnac11-2;zmexpb15-3* and *zmnac29-1;zmexpb15-3* double mutants and the *zmnac11-2;zmnac29-1;zmexpb15-3* triple mutant were investigated. The three double mutants were similar in HKW to the segregated *zmexpb15-3* single mutant, and also to *zmnac11-2* and *zmnac29-1* single mutants (Fig. 7a, b). Based on our findings that *ZmNAC11* and *ZmNAC29* could bind to *ZmEXPB15* and activate its transcription, we propose that *ZmEXPB15* may work in a common pathway with *ZmNAC11* and *ZmNAC29* in controlling kernel weight. This hypothesis is supported by the observation that *ZmEXPB15* expression was significantly decreased in *zmnac11* and *zmnac29* mutants (Fig. 7c), whereas the expression of *ZmNAC11* or *ZmNAC29* was not obviously altered in *zmexpb15* mutant (Supplementary Fig. 14a, b). However, *zmnac11;zmnac29;zmexpb15* triple mutants had even lower HKW, compared to the single and double mutants (Fig. 7a, b), suggesting that additional players are involved in the regulation of *ZmEXPB15* by *ZmNAC11/ZmNAC29*. Alternatively, the two NAC TFs and *ZmEXPB15* might also have partially independent roles in control of kernel weight. Taken together, we propose that *ZmEXPB15* may act, at least in part, in a common genetic pathway with *ZmNAC11* and *ZmNAC29* to control kernel weight (Fig. 7d).

## Discussion

Kernel size and weight are determined by the coordinated growth of embryo, endosperm, and maternal tissues. Current studies of maize kernel development mostly focus on the endosperm and embryo growth, and little is known about the control of the nucellus, a maternal tissue. This study provides insights into the regulation of

nucellus elimination and its impact on early endosperm development and consequently kernel-related traits.

Maize kernel size and weight are largely determined by the early development of the endosperm, in coordination with genetically distinct embryo and nucellus tissues during a short period around ~8 DAP[22,32]. Early endosperm development is accompanied by nucellus elimination though PCD[33,34], however, knowledge of key genes and mechanisms coordinating these processes in maize is lacking. We identified *ZmEXPB15* as the candidate gene of a hundred kernel weight QTL (*HKW9*), and show that it is expressed specifically in the nucellus from 2 to 8 DAP (Fig. 3a–c), coincident with the nucellus elimination period. A role of *ZmEXPB15* in the regulation of nucellus elimination is supported by the delayed appearance of TUNEL signal and lower Evan's blue staining in the small-kernel lines with lower *ZmEXPB15* expression (HKW^V671 and *zmexpb15* mutant) (Fig. 4j and Supplementary Fig. 8). This delay is associated with downregulation of PCD-related genes (Fig. 4k). It has been proposed that nucellus and endosperm development are antagonistic in Arabidopsis[11], as the nucellus degenerates in order to reallocate nutrients and space for endosperm development[3]. Consistent with these hypotheses, we observed that the endosperms of small-kernel lines were smaller as early as 3 DAP when the nucellus PCD was delayed, and the reduction in size became more severe at 12 DAP (Fig. 4 and Supplementary Fig. 8). These observations suggest that the reduced endosperm development in small-kernel lines is directly related to the difference in the timing of nucellus PCD, shedding light on the interaction between endosperm and maternal tissues in maize. Meanwhile, *ZmEXPB15* also promotes nucellus cell expansion which is in line with its protein localization and property, leading to a larger nucellus volume. The dual role of *ZmEXPB15* on nucellus cell expansion and elimination opens a window to understand

how the nucellus tissue growth coordinates with endosperm development. Our findings reveal a key role of *ZmEXPB15* in early endosperm development through promoting nucellus elimination and cell expansion, filling in a knowledge gap in the molecular coordination between endosperm and maternal tissues in maize.

Only a small number of genes are known to play important roles during early kernel development from fertilization to ~15 DAP in maize[23]. Most of these genes are mainly expressed after 6 DAP when the endosperm initiates, therefore their roles in kernel development are mostly through modulating early endosperm mitotic activity, or the differentiation and function of different endosperm cell types. To our knowledge, *ZmEXPB15* is the only gene showing a nucellus-specific expression pattern in maize, and it regulates early kernel development through a distinct process, promoting nucellus PCD and elimination. At early stages of cereal seed development, the nucellus is the first maternal tissue to degenerate, and allows the remobilization of cellular contents for embryo and endosperm development[13]. The identification of *ZmEXPB15* function highlights the importance of nucellus in early kernel development. *ZmEXPB15* encodes an expansin protein, which participates in various biological processes by affecting the loosening of the cell wall[35,36]. We found the ZmEXPB15 protein localized in cell wall as expected, which explains its role in nucellus cell expansion (Figs. 3d, 4a–c). Our main finding on its role in nucellus PCD process opens a window on expansin protein function, which might be related to the cytoplasm and nucleus localization of the GFP-ZmEXPB15 fusion protein (Fig. 3d). Further studies are required to elucidate the underlying mechanism on how *ZmEXPB15* triggers the initiation of PCD process.

The identification of two nucellus-enriched NAC TFs, *ZmNAC11* and *ZmNAC29* as candidate upstream regulators of *ZmEXPB15*, deepens our understanding on the molecular function of *ZmEXPB15*. Our results suggest that *ZmNAC11* and *ZmNAC29* may bind to the *ZmEXPB15* promoter to activate expression (Fig. 5d, e), and act epistatically with *ZmEXPB15* in regulating kernel size by promoting nucellus PCD (Fig. 7d). *zmnac11;zmnac29;zmexpb15* triple mutant had more severe kernel defects compared to the single or double mutants (Fig. 7a, b), suggesting that the two NAC TFs regulate additional kernel related genes. This hypothesis is supported by their broader expression, not only in nucellus but also in endosperm (Fig. 5c). Our discovery of the ZmNAC11/ZmNAC29-ZmEXPB15 module that triggers nucellus PCD and promotes its elimination to accommodate endosperm development uncovers a previously unknown mechanism regulating early kernel development in maize. Recently, a NAC-expansin regulatory module, NAC25/NAC1L-EXPA2, was found to regulate endosperm cell expansion during seed germination, and to control the seed-to-seedling transition in Arabidopsis. *NAC25/NAC1L* directly activates *EXPA2* expression specifically in endosperm, and promotes cell expansion[37]. In our study, the ZmNAC11/ZmNAC29-ZmEXPB15 module functions specifically in nucellus, to promote PCD process and cell elimination, and also promote its cell expansion revealing a unique function of the maize NAC-EXPANSIN module in kernel development.

Most kernel trait-related genes have been isolated from classical mutants with severe defects such as small or shrunken kernels, and are not directly applicable in breeding[5]. Our isolation of the maize kernel trait-related gene *ZmEXPB15* in this study was based on a hundred kernel weight (HKW) QTL, *HKW9*. We show that *ZmEXPB15* controls kernel size and weight, providing a resource to optimize maize kernel traits. To support its potential breeding value, we identified an elite haplotype (*Hap1*) for *ZmEXPB15*, which has increased *ZmEXPB15* expression and HKW (Fig. 1j–l), and can be applied in molecular breeding for large kernel lines through marker-assisted selection (MAS). The large-kernel NIL, HKW9[Mc], increased hundred kernel weight, kernel length and width, without obvious alteration in other agronomic traits, including starch content (Fig. 1a–e, Supplementary Fig. 1). Thus, this germplasm could be used as a large-kernel donor to

improve other inbreds. In addition, *ZmEXPB15* overexpression increased kernel weight and size, and this increase was consistent in hybrid combinations with different inbred lines, without an obvious negative effect in other ear traits. Therefore, our studies reveal the potential application of *ZmEXPB15* in molecular breeding to generate large-kernel germplasm to improve crop yield-components.

## Methods

### Construction of the NIL lines
Two elite maize inbred lines, Mc and V671, that develop small and large kernels, respectively, were used for mapping hundred kernel weight (HKW) QTL. Using $F_{2:3}$ families, a major QTL (*HKW9*) on chromosome 9 region (bin 9.03–9.04) was detected, and accounted for more than 10% of phenotypic variation[19]. To construct near-isogenic lines (NILs), Mc was used as the recurrent parent, and the HKW9 allele from V671 was introduced into Mc background through 4 generations of back-crossing followed by self-crossing. This resulted in two NILs, HKW9[Mc] and HKW9[V671], harboring the *HKW9* locus from Mc and V671, respectively.

### Kernel trait measurement
Two near-isogenic lines (NILs) harboring the major hundred kernel weight QTL (*HKW9*) locus from Mc and V671, HKW9[Mc] and HKW9[V671], were planted at different locations using a randomized block design with three replicates. For different transgenic materials (such as KO, OE, *nac*, *nac-exp*), stable generations of $F_3$ or $F_4$ were used for phenotypic identification. Well-pollinated ears in the middle of each row were selected to measure kernel-related traits, including the kernel length and width, and the hundred kernel weight. Microsoft Excel 2010 was used for significance test and analysis. Ear and kernel phenotype images were captured using a Nikon D7100 camera. Box plot and histogram was produce using GraphPad Prism 8 software. The kernel length and width were scored using a yield-trait scorer (YTS-MKT, GREENPHENO). One hundred kernels from the middle part of the ears were selected for measurement of hundred kernel weight. Kernel starch content was measured using the megazyme total starch assay kit (Catalog: K-TSTA-50A) according to the manufacturer's instructions.

### Cytological observation of developing kernels
For tissue sections for endosperm and nucellus area measurements, we made 1-mm median longitudinal sections of kernels with 0-12 DAP, fixed overnight in 4% (v/v) paraformaldehyde (Sigma–Aldrich), dehydrated in an ethanol gradient series (30%, 50%, 70%, 85%, 95%, and 100% ethanol), and embedded in Paraplast Plus (Sigma–Aldrich). The sample blocks were sectioned into 8-μm slices, and the longitudinal median sections were selected for measurement according to the following criteria for different developmental stage. 0 DAP selects the maximum area of the ovules, 2-3 DAP selects the maximum area of the endosperm, and for the kernels at 4 DAP and later stages, the maximum area of embryos was used for the reference, since the entire endosperm was too large to judge accurately. Finally stain with 0.5% (w/v) toluidine blue. Images were captured under the microscope (Nikon eclipse 80i).

For semi-thin sections, the kernels were cut longitudinally, leaving 1/5 of the middle part about 2 mm thick. The samples were immediately fixed in 50% FAA solution, resin embedding was performed with the kit Technovit 7100 (Kulzer, Germany), stained with 0.5% (w/v) toluidine blue solution after section, and photographed with Nikon camera (Nikon eclipse 80i). Finally, the cell size was measured and calculated by Image J 1.50i.

For scanning electron microscopy observation, endosperms at 16 DAP were cut into 2-mm³ pieces. Fresh tissues were fixed overnight in 2.5% (w/v) glutaraldehyde in sodium phosphate buffer (0.1 M, pH 7.2) at 4 °C for 12 h. Samples were dehydrated in a graded series of ethanol,

and then dried to the critical point, sputter-coated with gold in an E-100 ion sputtering process, and observed with a scanning electron microscope (6390LV; JEOL, Tokyo, Japan).

## Phylogenetic analysis

The ZmEXPB15 sequences from HKW9[Mc] and HKW9[V671] were aligned using ClustalX software with the default parameters. The phylogenetic tree for multiple NAC transcription factors of maize was constructed using the MEGA program version 7 (www.megasoftware.net) and the neighbor-joining method, whose parameters are Poisson correction, pairwise deletion, and boot-strap (500 replicates; random seed).

## Genetic transformation

To create knockout mutants for ZmEXPB15, ZmNAC11 and ZmNAC29, the guide RNA sequences (Supplementary Figs. 5c, 11b) against target genes were designed at the website of CRISPR-P 2.0 (http://crispr.hzau.edu.cn/CRISPR2/)[38], then were cloned into a CRISPR/Cas9 plant expression vector pCPB-ZmUbi-hspCas9[39], followed by the agrobacterium-mediated transformation into the maize inbred line C01. The primers used for the transgenic genotype detection were listed in Supplementary Data 3.

To overexpress *ZmEXPB15*, the full CDS of the *ZmEXPB15* copy from the large-kernel NIL, HKW9[Mc], driven by the ubiquitin promoter was ligated to the binary vector pCAMBIA3301 and then transformed into the maize inbred line B104. To construct *proZmEXPB15:ZmEXPB15*-GFP, the genomic fragment including 1.5 kb promoter, full CDS and eGFP-fused gene was amplified and inserted into the pCAMBIA3301 entry vector, and then transformed into the maize inbred line B104. The Basta (bar) gene was amplified to distinguish the transgenic and non-transgenic plants. The primers used for the transgenic genotype detection were listed in Supplementary Data 3.

## RT-qPCR assay

Immature (0-12 DAP) kernels from three independent ears were dissected and quickly frozen in liquid nitrogen, and were ground into a fine powder. Total RNA was extracted with TRIzol reagent (Thermo Fisher Scientific) according to the manufacturer's instruction, and was treated with DNaseI digestion. For the reverse transcription, 1 μg of total RNA was used in each 20 μL reaction and the first cDNA strand was synthesized using Super-ScriptII (Invitrogen). The qRT-PCR analysis was carried out with three independent RNA samples using a Bio-Rad CFX96 Touch Real-Time PCR detection system using *Actin* as an internal control, and the relative expression of the target transcripts was calculated using the $2^{-\Delta\Delta CT}$ method[40]. The primers used for the amplicons were listed in Supplementary Data 3.

## RNA in situ hybridization

Immature C01 kernels at 0-12 DAP were collected and fixed in FAA solution containing 5% formalin (v/v), 50% ethanol (v/v), and 5% acetic acid (v/v) for 16 h at 4 °C, and then replaced with 70% ethanol twice and dehydrated with an ethanol series, followed by the substitution with xylene and embedding in Paraplast Plus (Sigma–Aldrich). For making sense and antisense RNA probes, gene-specific primers (listed in Supplementary Data 3) were used to amplify *ZmEXPB15, ZmNAC11* and *ZmNAC29*. The amplified products were cloned into pSPT18 (Roche) and linearized with *Hind* III or *EcoR* I. The sense and antisense probes were separately synthesized by RNA polymerase using SP6 or T7 primer with digoxigenin-labelled UTP. RNA hybridization and immunologic detection of the hybridized probes were performed according to Greb[41], with the addition of 8% polyvinyl alcohol to the detection buffer to minimize diffusion of the reaction products. Slides were exposed for 12–15 h before mounting and imaging, and were visualized under a microscope (Nikon eclipse 80i).

## Confocal microscopy observation

Kernels at 2-8 DAP from *proZmEXPB15:ZmEXPB15*-GFP plant were cut longitudinally with a thickness of about 1 mm and observed under a confocal microscope (Olympus FV1000) with an excitation wavelength at 488 nm.

## Terminal deoxynucleotidyl transferase dUTP nick-end labeling assay

The paraplast fixed developing kernels at 0-6 DAP were from above used for the toluidine blue staining. The paraplastp was removed by the axylol treatment, and the sections were hydrated with an ethanol series and treated with proteinase K in PBS. The endogenous peroxidase activity was quenched by the treatment with $H_2O_2$. The sections were incubated at 37 °C for 60 min in the presence of TdT using the TUNEL apoptosis detection kit (DeadEnd[TM] fluorometric TUNEL system; Promega). The TUNEL-positive signal was detected using Dead-End[TM] Fluorometric TUNEL System (Promega) following the manufacturer's instructions.

## Evan's blue staining

The nucellus tissue from different days after pollination was dyed in Evan's blue solution and shaken for 20 min to ensure that all tissues were exposed to Evan's blue dye solution. Rinse thoroughly with deionized water until the unbound dye washes away from the surface, then photograph. Quantitative Evan's blue staining was performed by extracting Evan's blue dye from stained nucellus tissue, in which 100 mg of tissue was thoroughly ground with 1 mL 1% sodium dodecyl sulfate (SDS) cell lysis buffer and sieved. Directly transfer 250 μL of supernatant to a 96-well microtiter plate. Measure the optical density at 600 nm spectrophotometrically using a plate reader by taking 1% SDS as blank. The concentration of Evan's blue can be estimated using a reference standard curve[42].

## RNA-seq analysis

Total RNA of HKW9[Mc] and HKW9[V671] were extracted from 6-DAP kernels using TRIzol reagent, followed by cleanup and DNase I treatment using an RNeasy Mini Kit (Qiagen). Three independent biological replicates from different ears were performed. cDNA libraries were constructed following the standard Illumina protocol and sequenced on the Illumina NovaSeq platform by Novogene. The sequence reads were trimmed using the tool Trimmomatic (v.0.33; http://www.usadellab.org/cms/?page5trimmomatic) and were mapped to the B73 RefGen_v4.34 reference genome using the programHISAT2 (v.2.1.0; http://ccb.jhu.edu/software/hisat2/faq.shtml). The program StringTie (v.1.3.3b) was employed to reconstruct the transcripts and to estimate gene expression levels[43]. The program HTSeq (v.0.6.1; https://pypi.org/project/HTSeq/) was used to count the number of reads per gene. The differentially expressed genes (DEGs) were identified by DEseq2 (http://www.bioconductor.org/packages/devel/bioc/html/DESeq2.html) by a *p*-value of <0.001, absolute value of log$_2$[fold change] > 0.5. DEGs are listed in Supplementary Data 1.

## Transient expression assays in maize protoplast

For promoter activation assay, the full coding sequences of ZmNAC11 and ZmNAC29 driven by the 35 S promoter were cloned into the effector vector pGreenII 62-SK, and different truncated promoter of the *ZmEXPB15* from HKW9[Mc] and HKW9[V671] were cloned into the reporter vector pGreenII 0800-LUC. The empty pGreenII 62-SK vector was used as control. The transient dual-luciferase assays were performed in the maize protoplasts collected from the leaves of 2-week-old etiolated seedling of inbred line B73. The luciferase signal was detected using dual-luciferase assay reagents (Promega) following the manufacturer's instructions. The ratio of LUC/REN activity was measured using multi-mode microplate reader (SpectraMax i3x). Primers are listed in Supplementary Data 3.

## Recombinant protein expression and purification

The full coding sequences of ZmNAC11 and ZmNAC29 were cloned into the recombinant protein expression vectors pET-28a-SUMO vector (Novagen), and were introduced into *E.coli* BL21 (DE3) strain. The recombinant proteins, His-SUMO-ZmNAC11 and His-SUMO-ZmNAC29, were induced with 0.2 mM isopropyl-β-D-thiogalactoside (IPTG) for 16 h at 16 °C, and were purified with columns equipped with Ni$^{2+}$ affinity resin (GE Healthcare). Primers are listed in Supplementary Data 3.

## Electrophoretic mobility shift assay

Oligonucleotide probes from −299 to −249 upstream of the *ZmEXPB15* promoters from HKW9$^{Mc}$ and HKW9$^{V671}$ were synthesized and labeled with the Pierce Biotin 3′ End DNA Labeling Kit. The purified recombinant His-SUMO-ZmNAC11 and His-SUMO-ZmNAC29 proteins were used to detect the binding affinity with the biotin-labeled probes by the LightShift™ Chemiluminescent EMSA kit (Thermo Fisher Scientific) according to the standard manufacturer's manual. Primers are listed in Supplementary Data 3.

## Reporting summary

Further information on research design is available in the Nature Research Reporting Summary linked to this article.

## Data availability

The plant materials generated in this study are available from the corresponding authors upon request. The RNA-seq datasets are available from the NCBI Sequence Read Archive under the accession numbers PRJNA756197 and SRR15525330-SRR15525335. Source data are provided with this paper.

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

## Acknowledgements

This work was supported by grant from the National Key Research and Development Program of China (2021YFF1000500, F.Q.) and the National Natural Science Foundation of China (31971951, F.Q. and 31771796, F.Q.). The computations in this paper were run on the bioinformatics computing platform of the National Key Laboratory of Crop Genetic Improvement, Huazhong Agricultural University.

## Author contributions

F.Q. and F.Y. conceived and designed the project. Q.S., Y.L., A.H., W.Z., Q.N., Z.T., K.L. and L.M. performed the experiments. D.G. and W.Z. designed the CRISPR sgRNAs and prepared the constructs for maize transformation. H.Z. and Z.T. analysed the RNA-seq data. Q.S., F.Y., Z.Z. and D.J. analysed the data and wrote the manuscript.

## Competing interests

The authors declare no competing interests.
