## [Peer Review File · Nature Communications]

A NAC-EXPANSIN module enhances maize kernel size by controlling nucellus eliminationReviewers' Comments:

Reviewer #1:

Remarks to the Author:

This study established a new Nac-Expansin pathway regulating maize kernel development through promoting nucellus degeneration, and demonstrated a potential application of ZmEXPB15 (HKW9Mc) in molecular breeding to increase maize yield. The manuscript is well organized and clearly written, the data is of high quality and convincing, and the findings are of high novelty and significance. It was a pleasure to read.

I have only a few minor comments for the authors:

1. For the general audience, please briefly explain the origin, development, and the role of nucellus in kernel (seed) development;
2. Is it possible to describe the construction process of the two NILs lines, HKW9Mc and HKW9Mc (which generation, what background in etc.);
3. I am not sure why the hundred kernel weight of HKW9Mc/HKW9V671 and HKW9V671/HKW9Mc F2 kernels was similar to that of HKW9Mc/HKW9Mc F2 kernels and smaller than that of HKW9V671/HKW9V671 F2 kernels (Fig. 1g). What does this observation implies? Please elaborate.
4. The map position of KKW9 needs to be shown;
5. Is the 73 kb region flanked by umc2370 and bnlg1209 completely sequenced for both the parental lines? Any gap?
6. Are the SNPs in the promoter associated with other kernel traits besides hundred kernel weight?
7. Does the 4-bp deletion in the last exon plays a role in kernel size regulation?
8. For molecular breeding, have the authors designed some usable markers that can be easily applied?

Reviewer #2:

Remarks to the Author:

The article "A NAC-EXPANSIN module enhances maize kernel size by controlling nucellus degeneration" by Sun Q and colleagues describes the contribution of ZmEXPB15 and two regulators, ZmNAC11 and ZmNAC29, to kernel size and weight.

The ZmEXPB15 gene, encoding an expansin, was identified through the study, in two near isogenic lines (NILs) of a QTL related to hundred-kernels weight. The authors show that ZmEXPB15 overexpression leads to bigger and heavier kernels, and loss-of-function to the opposite phenotypes. ZmEXPB15 is shown to be specifically expressed and to act in the nucellus. Two transcription factors, ZmNAC11 and ZmNAC29, are shown to bind and activate ZmEXPB15 promoter. Both ZmNAC11 and ZmNAC29 loss-of-function show decrease in kernel size and weight. ZmNAC11 and ZmNAC29 are shown to be involved in nucellus cell death and endosperm growth. ZmEXPB15 is also proposed to regulate these processes even if not directly shown (see comment 1 below). Finally, the authors proposed a model in which ZmNAC11 and ZmNAC29 regulate kernel weight and size by enhancing

nucellus cell death and thus endosperm growth, partially through the activation of ZmEXPB15 expression.

The article is written in a good English and is easy to follow. The different parts (introduction, methods, results, and discussion) meet the standards of the field and adequate references are given in most cases (exception for comment 6).

The article presents an original story, showing new genes involved in the regulation of kernel size and weight. Unlike most of the known genes regulating these traits that act either in the endosperm or the embryo, these genes act in the nucellus, highlighting the importance of nucellus-endosperm interaction for proper kernel growth. As discussed in the paper, similar interaction has already been shown in other species such as rice and Arabidopsis, but with other genes and never in maize.

Another strength of the article is its potential agronomic outcome, especially the overexpression of ZmEXPB15, which increases kernel weight and size with no negative effect on ear length.

The results on quantification of kernel traits are very robust with relevant statistical analyses that strongly support the conclusions. The conclusion on ZmEXPB15 regulation by ZmNAC11 and ZmNAC29 are also strongly supported by the results presented. However, the conclusions on the developmental origin of the kernel weight and size phenotypes are a bit weakly supported, especially regarding cell death (see comment 1, 6 and 7) and should be fleshed out. However, I think that the paper is of high quality and if the authors can address the following comments, I recommend its publication in Nature Communications.

Major comments

1) In the abstract it is stated that ZmEXPB15 positively controls kernel size and weight by promoting nucellus degeneration. However, this is only based on the study of the two NILs that have also several differentially expressed genes. Did you also observe a delay in nucellus cell death and endosperm expansion in the ZmEXPB15 KO lines?

2) Details must be provided, at least in the material and method, on how the two NILs have been generated.

In ref. 18 (Liu, Y. et al. 2014), the Mc line is described as giving smaller kernels than the V671 line. Here it is the opposite: the HKW9Mc locus gives big kernels and the HKW9V671 locus gives small kernels. Why?

In line 82, the sentence is a bit confusing to me. What do you mean by "were increased"? Does this mean that the difference between the two NILs was smaller in the past?

3) Line 105: Two candidate genes are strongly upregulated in the kernel. However, you only focused on ZmEXPB15. What about the second gene upregulated? It also encodes an expansin (EXPB14). Is there any change in its DNA sequence between the two NILs? Is it expressed in the nucellus too? Do you think it could be redundant with ZmEXPB15?

4) The authors paid a lot of attention on the difference in ZmEXP15 expression level.

However, the ZmEXPB15 proteins are significantly different in the two NILs, due to a 4pb deletion and a shift in the ORF in HKW9V671. Where are the conserved domains of ZmEXPB15? Is the protein function potentially altered in HKW9V671?

5) Line 214: "GFP signal was enriched at the cell periphery, consistent with its predicted cell wall localization". The protein localisation in the cell walls in Fig. 3D is not clear to me. Did the authors try to counter stain the cell walls or to perform a plasmolysis to assess the cell wall localization?

6) Line 252: the authors studied the expression level of genes that are claimed to be PCD-related. Could the authors provide a reference showing that these genes are PCD-related?

7) Based on Figure 4, the difference between the two NILs in cell death and in endosperm expansion are very distinct in timing, suggesting they are not related. Indeed, in Fig 4c, difference in TUNEL signal between the two NILs can only be seen at early stages and at 6DAP no difference is observed. However, at 6 DAP, no difference between the two NIL can be seen in the endosperm/nucellus ratio, the difference appears only at later stage (Fig 4b). This suggests that these two processes are not directly related because separated in time.

It could come from the fact that TUNEL is not the best proxy of cell death, as it only shows the presence of DNA fragmentation, which can happen very early in PCD. The use of other cell death markers such as Evans blue staining, FDA staining or cytological description on FFPE section or ultrastructural analyses by TEM would maybe give a better correlation between the difference in cell death and in endosperm growth and might be informative to understand more about the developmental origin of the phenotype described.

Minor comments:

Figure 7: indicate which zmexpb15 mutant you used.

Line 95: change smaller by higher

Sup. Figure 1: change seeds by kernels.

Sup Figure 10: change c in the text by b.

Reviewer #3:

Remarks to the Author:

Sun et al., deciphered the functional implication of expansin gene (ZmEXPB15) in mediating nucellus degeneration, affecting endosperm growth and thus grain weight. In this study authors clearly demonstrated that ZmNAC11 and ZmNAC29 directly bind to ZmEXPB15. The knockout lines of these transcription factors (ZmNAC11 and ZmNAC29), ZmEXPB15, and triple mutants exhibit delayed programmed cell death (PCD) of maternal tissue nucellar projection, wherein endosperm development is impacted with reduced kernel size and weight. The proposed work identified the regulation of the NAC-Expansin module but does

not offer mechanistic insights. Also, the authors speculated that haplotype1 targeted for large-kernel NIL, HKWMc might be of use to increase grain weight in the breeding programs but do not offer direct evidence of breeding application in the manuscript.

Revealing further clarity will help to pinpoint the role of ZmEXPB15 in mediating nucellus-endosperm growth dynamics and its impact on grain weight:

1. Although the Authors showed that ZmNAC11, ZmNAC29, and ZmEXPB15 individual knock-out lines exhibit lowered hundred kernel weight, more so in the triple mutant exhibit, the exact molecular function of ZmEXPB15 in mediating PCD of nucellus through potential cell wall loosening or cell wall degeneration not shown. Also, there were many proteases reported to be differentially regulated in these mutants. Bringing further clarity between the NAC-Expansin module and proteases will be helpful.
2. Authors claimed that ZmEXPB15 contributes to the yield improvement of maize. To substantiate this claim yield data is a pre-requisite (plot yield) for CRISPR-Cas9 mutants and transgenic overexpression lines.
3. Discussion is rather repetitive with results, offering a clear insight of molecular function of expansin with the onset of PCD will be helpful.

Reviewer #4:

Remarks to the Author:

The manuscript "A NAC-EXPANSIN module enhances maize kernel size by controlling nucellus degeneration" by Qin Sun et al. describes the role of one EXPANSIN and two NAC genes in nucellus elimination during early maize kernel development. The authors identify the ZMEXPB15 gene as candidate for the HKW9 QTL, which affects kernel size and weight. Genetic and expression analyses show that this EXPANSIN is involved in the elimination of the nucellus and, as a consequence, affects the development of the endosperm and the overall kernel as well. Finally, the manuscript presents data suggesting that the transcription factors NAC11 and 29 work upstream ZMEXPB15 to facilitate the elimination of the nucellus.

The results are novel and interesting both in the of basic and applied research sphere. The manuscript is clear and well written.

Major points:

-My main concern regards the interpretation of the mutant phenotypes. The authors should check if ZMEXPB15 and NAC11 and 29 are involved in the expansion of the nucellus that follows fertilization, as the cell elimination phenotype might be only indirect. While some cells of the nucellus are eliminated, the others have to follow the expansion of the rest of the seed. Regardless of which hypothesis will turn out to be true, a more thorough phenotypic analysis has to be conducted to better image the processes of cell expansion and elimination.

-I believe that results from ZMEXPB15 over-expression lines would be meaningful only if the authors clearly show an effect on nucellus development.

-Fig 4 and Fig6: the results obtained by measuring nucellus and endosperm areas strongly depend on the section used. How do the authors pick their sections? How do they make

sure to have longitudinal-mid sections?

-Fig 4: The authors should have performed the same analyses also with Zmexp15 CRISPR lines.

-Data on the direct binding of NAC 11 and 29 on the ZMEXPB15 promoter are not conclusive. A Chip experiment is necessary to claim direct binding. I understand that it is not an easy experiment, therefore I would ask the authors to adjust model and text accordingly. Furthermore, I do not see signs of competition in some of the EMSA analyses.

-The expression pattern of NAC 11 and 29 should be more carefully characterized by RNA in situ hybridization experiments.

Minor Points:

-The process of nucellus death is now referred to as "cell elimination" because it leads to the full degeneration of the cell including the cell wall.

-Line 112: The data presented do not demonstrate that the differential expression is due to the promoter SNPs.

We would like to sincerely thank all reviewers for their constructive criticisms and suggestions. These comments are of great help to improve the manuscript. We have conducted all the experiments as suggested and addressed all the comments, and the manuscript has been revised accordingly. Our responses to reviewers' comments are in blue below, and in the revised manuscript.

Reviewer #1 (Remarks to the Author):

This study established a new Nac-Expansin pathway regulating maize kernel development through promoting nucellus degeneration, and demonstrated a potential application of ZmEXPB15 (HKW9Mc) in molecular breeding to increase maize yield. The manuscript is well organized and clearly written, the data is of high quality and convincing, and the findings are of high novelty and significance. It was a pleasure to read.

I have only a few minor comments for the authors:

1. For the general audience, please briefly explain the origin, development, and the role of nucellus in kernel (seed) development;

Response 1.1 Thanks for your positive comments and constructive criticisms. We introduced the nucellus in more detail in the Introduction.

“The maternal nucellar tissue, originating from the ovule, acts as a bridge for apoplasmic intercellular exchange between the mother plant and its developing offspring. It also provides an environment for the developing zygote and filial tissue^{1, 2}.” (lines 49-52)

“At an early stage, the maternal nucellar tissue degenerates by programmed cell death (PCD), and the contents of the dying cells are re-mobilized to feed the developing embryo and endosperm^{3, 4}.” (lines 56-58)

2. Is it possible to describe the construction process of the two NILs lines, HKW9Mc and HKW9Mc (which generation, what background in etc.);

Response 1.2 Thank the reviewer for the reminder. We introduced the construction process of the two NIL lines in more detail in the Methods. **(lines 597-605)**

“Two elite maize inbred lines, Mc and V671, that develop small and large kernels, respectively, were used for mapping hundred kernel weight (HKW) QTL. Using F_{2:3} families, a major QTL (HKW9) on chromosome 9 (bin 9.03-9.04) was detected, and accounted for more than 10% of phenotypic variation⁵. To construct near-isogenic lines (NILs), Mc was used as the recurrent parent, and the HKW9 allele from V671 was introduced into Mc background through 4 generations of backcrossing followed by self-crossing. This resulted in two NILs, HKW9^{Mc} and HKW9^{V671}, harboring the

HKW9 locus from Mc and V671, respectively.”

3. I am not sure why the hundred kernel weight of HKW9Mc/HKW9V671 and HKW9V671/HKW9Mc F₂ kernels was similar to that of HKW9Mc/HKW9Mc F₂ kernels and smaller than that of HKW9V671/HKW9V671 F₂ kernels (Fig. 1g). What does this observation implies? Please elaborate.

Response 1.3 Thanks for your question. We apologize there was a mistake in the manuscript, that is “the hundred kernel weight of HKW9^{Mc}/HKW9^{V671} and HKW9^{V671}/HKW9^{Mc} F₂ kernels was similar to that of HKW9^{Mc}/HKW9^{Mc} F₂ kernels and ‘smaller’ than that of HKW9^{V671}/HKW9^{V671} F₂ kernels” It should be ‘larger’ and we corrected the mistake (*lines 99-102*).

In the F₁ generation, the hundred kernel weight of HKW9^{Mc}/HKW9^{V671} (M/V) with ‘M’ as the maternal genotype was larger than that of HKW9^{V671}/HKW9^{Mc} (V/M). This result indicates that *ZmEXPB15* functions maternally. The maternal effect is determined by the maternal genotype rather than the progeny genotype. The M/V and V/M F₂ kernels both have the heterozygous maternal genotypes, and thus show the same large-kernel phenotype. Please see the diagram below.

Figure legend. Diagram of the different genotypes developed for the maternal effect test in Fig. 1g. M/M, M/V, V/M and V/V represent the genotype of the individual, corresponding to HKW9^{Mc}/HKW9^{Mc}, HKW9^{Mc}/HKW9^{V671}, HKW9^{V671}/HKW9^{Mc} and HKW9^{V671}/HKW9^{V671}, respectively.

4. The map position of HKW9 needs to be shown;

Response 1.4 The map position of the QTL *HKW9* was described in an article previously published in our laboratory⁵. Based on the genetic information from Liu et al. (2014), we constructed the map position of *HKW9*, which mapped across the centromere (please see the figure *a* below). The two NILs described earlier, HKW9^{Mc} and HKW9^{V671}, were used in Bulk Segregant Analysis to identify a single peak of ~70 Mb which overlapped with the QTL *HKW9* (please see the figure *b* below). We tried to narrow down the *HKW9* mapping region using more markers, but could not, indicating that the candidate region had low recombination. Given that the *HKW9* locus was close to the centromere, which makes fine mapping difficult, we isolated the causal gene in combination with transcriptome sequencing and association study. Since the *HKW9* map region was not further narrowed down and was reported

previously, we did not show it as a figure.

Figure legend. a. The molecular linkage map of the QTL *HKW9* based on 22 SSR markers on chromosome 9 using the raw data from the article published by Liu et al 2014⁵. The linkage map of total length 1,351.7 cM across the centromere with an average 5.28 cM interval between adjacent markers, was constructed by Mapmaker/EXP V3.0 software with “error detection on” at logarithm of odds (LOD) threshold >3.72. **b.** The distribution of Delta SNP-index on chromosome 9 was detected by Bulk Segregant Analysis (BSA) using an F₃ population derived from the two NILs, *HKW9*^{Mc}/*HKW9*^{V671}. Delta SNP-index was obtained by the SNP-index of small-kernel pool minus the SNP-index of large-kernel pool. The red curve is the mean value of SNP-index, green is the threshold line of 95% confidence level, and orange is the 99% confidence level.

5. Is the 73 kb region flanked by *umc2370* and *bnlg1209* completely sequenced for both the parental lines? Any gap?

Response 1.5 Actually, there are 73 Mb between *umc2370* and *bnlg1209*. Since the region is large and across the centromere, we did not sequence it, but tried to narrow down the region (please refer to the **Response 1.4**).

6. Are the SNPs in the promoter associated with other kernel traits besides hundred kernel weight?

Response 1.6 Thank the reviewer for the valuable suggestion. We did association analysis of these 5 SNPs with other kernel traits, but they were not significantly associated with kernel length, kernel width or kernel thickness in an association population of 220 lines. In addition, the expression level of *ZmEXPB15* was not correlated with these kernel traits. Please see the results in the table below.

	Kernel length	Kernel width	Kernel thickness
Association analysis of genotype and phenotype (* p < 0.026)	0.040	0.039	0.037
Correlation analysis of expression level and phenotype (r < 0.3 irrelevant)	0.19	0.17	0.076

7. Does the 4-bp deletion in the last exon plays a role in kernel size regulation?

Response 1.7 Thanks for your question. The 4-bp deletion in the last exon of *ZmEXPB15* caused a frame shift of the last 60 amino acids and an addition of 24 residues at the C-terminus (**updated Supplementary Fig. 3**). Based on the alignment of *ZmEXPB15* sequences across different species, the last 60-AA region is conserved

(please see the alignment below, red box), therefore, we could not rule out the possibility that this 4-bp deletion might interrupt the function of *ZmEXPB15*. A direct proof for this possibility would be a complementation test using this 4-bp deletion version of *ZmEXPB15* protein in *zmexpb15* mutant. However, maize transformation is limited to few specific genetic backgrounds, and takes ~8 months, and another 8 months (2 generations) are needed to introgress the transgene into the *zmexpb15* mutant for the complementation. We regret that we are not able to provide this data in the limited time. Meanwhile, we showed that the SNPs in the promoter correlate with the expression level of *ZmEXPB15* and also with the hundred kernel weight. Nevertheless, we have provided several lines of evidence showing that *ZmEXPB15* is responsible for the difference of HKW between the two NILs, HKW9^{V671} and HKW9^{Mc}, and that *ZmEXPB15* is an important gene controlling kernel weight and size.

Figure legend. Alignment of the amino acid sequences of *ZmEXPB15* and the homologues in other species. Black shaded amino acids represent the residues with 80% identity and gray ones indicate the similar residues. The red box indicates conservation of the last 60 amino acids.

8. For molecular breeding, have the authors designed some usable markers that can be easily applied?

Response 1.8 Yes, we developed one usable Kompetitive Allele-Specific PCR (KASP) marker based on the third SNP (T-A) significantly associated with hundred kernel weight. This marker (F₁/F₂/R) could differentiate the two haplotypes and can be easily applied for molecular breeding (please see the figure below).

Figure legend. Genotypes of an associated population of 100 lines were separated using the KASP marker, which contains two SNP-specific primers (F₁/F₂) and one universal primer (R). The green and blue dots represent the *Hap1* and *Hap2* genotypes, respectively. The gray and black dots represent negative control and undetermined genotypes, respectively.

Reviewer #2 (Remarks to the Author):

The article “A NAC-EXPANSIN module enhances maize kernel size by controlling nucellus degeneration” by Sun Q and colleagues describes the contribution of ZmEXPB15 and two regulators, ZmNAC11 and ZmNAC29, to kernel size and weight.

The ZmEXPB15 gene, encoding an expansin, was identified through the study, in two near isogenic lines (NILs) of a QTL related to hundred-kernel weight. The authors show that ZmEXPB15 overexpression leads to bigger and heavier kernels, and loss-of-function to the opposite phenotypes. ZmEXPB15 is shown to be specifically expressed and to act in the nucellus. Two transcription factors, ZmNAC11 and ZmNAC29, are shown to bind and activate ZmEXPB15 promoter. Both ZmNAC11 and ZmNAC29 loss-of-function show decrease in kernel size and weight. ZmNAC11 and ZmNAC29 are shown to be involved in nucellus cell death and endosperm growth. ZmEXPB15 is also proposed to regulate these processes even if not directly shown (see comment 1 below). Finally, the authors proposed a model in which ZmNAC11 and ZmNAC29 regulate kernel weight and size by enhancing nucellus cell death and thus endosperm growth, partially through the activation of ZmEXPB15 expression.

The article is written in a good English and is easy to follow. The different parts (introduction, methods, results, and discussion) meet the standards of the field and adequate references are given in most cases (exception for comment 6).

The article presents an original story, showing new genes involved in the regulation of kernel size and weight. Unlike most of the known genes regulating these traits that act

either in the endosperm or the embryo, these genes act in the nucellus, highlighting the important of nucellus-endosperm interaction for proper kernel growth. As discussed in the paper, similar interaction has already been shown in other species such as rice and Arabidopsis, but with other genes and never in maize.

Another strength of the article is its potential agronomic outcome, especially the overexpression of *ZmEXPB15*, which increases kernel weight and size with no negative effect on ear length.

The results on quantification of kernel traits are very robust with relevant statistical analyses that strongly support the conclusions. The conclusion on *ZmEXPB15* regulation by *ZmNAC11* and *ZmNAC29* are also strongly supported by the results presented. However, the conclusions on the developmental origin of the kernel weight and size phenotypes are a bit weakly supported, especially regarding cell death (see comment 1, 6 and 7) and should be fleshed out. However, I think that the paper is of high quality and if the authors can address the following comments, I recommend its publication in Nature Communications.

Major comments

1) In the abstract it is stated that *ZmEXPB15* positively controls kernel size and weight by promoting nucellus degeneration. However, this is only based on the study of the two NILs that have also several differentially expressed genes. Did you also observe a delay in nucellus cell death and endosperm expansion in the *ZmEXPB15* KO lines?

Response 2.1 Thanks to the reviewer for the positive comments and constructive suggestions. Following your suggestion, we investigated the *ZmEXPB15* KO lines and also observed a delay in nucellus cell death and endosperm expansion. To collect more data, we investigated the PCD process of nucellus in *ZmEXPB15* KO as well as OE lines and got consistent results, as shown in the **newly added Supplementary Fig. 8** (please also see the figure below). Briefly, TUNEL assays of the nucellus at 3 and 6 DAP (days after pollination) demonstrated that the KO lines showed an absent or weaker signal, whereas OE line showed an earlier and stronger signal, compared to the controls (**newly added Supplementary Fig. 8c, d**). Furthermore, Evan's blue staining which marks cell death showed an enhanced blue staining in the large-kernel line HKW9^{Mc} and OE line, demonstrating that increased *ZmEXPB15* expression could lead to enhanced nucellus cell death (**newly added Supplementary Fig. 8d, f**). Finally, the KO line had a higher ratio of nucellus to endosperm area due to the delay of nucellus PCD at 3 DAP, and the OE line showed the opposite, as expected (**newly added Supplementary Fig. 8b**). These results consistently demonstrate that *ZmEXPB15* promotes nucellus elimination.

Figure legend. *ZmEXPB15* promotes nucellus cell elimination. **a**, Ratio of nucellus (orange box) to endosperm area (green box) of 3-DAP kernels in *ZmEXPB15* knockout line (KO1) in comparison to the wild type (WT1) control in a C01 inbred background, and overexpression line (OE1) in comparison to the non-transgenic line (NT1) in a B104 inbred background. Longitudinal median sections with maximum area of the entire endosperm from more than 10 kernels for each line were examined and quantified. **b, c**, TUNEL assays of the nucellus at 3 and 6 DAP of KO1 in comparison to WT1 (**b**), and the OE1 in comparison to NT1 control (**c**). **d, e**, Evan's blue staining of young kernels of NILs HKW9^{Mc} and HKW9^{V671} at 4 and 6 DAP (**d**), and the kernels of OE1 in comparison to NT1 control at 2 and 4 DAP (**e**). The quantification was performed by measuring the blue dye absorbance at 600 nm. The values are shown as the means \pm s.d., and the significance is estimated by a two-tailed Student's *t* test, **P* < 0.05, ****P* < 0.001. Scale bar = 100 μ m in (**b, c**).

2) Details must be provided, at least in the material and method, on how the two NILs have been generated.

In ref. 18 (Liu, Y. et al. 2014), the Mc line is described as giving smaller kernels than the V671 line. Here it is the opposite: the HKW9^{Mc} locus gives big kernels and the HKW9^{V671} locus gives small kernels. Why?

Response 2.2.1 We thank the reviewer for the question. The QTL for hundred kernel weight (HKW) was originally isolated from two elite maize inbred lines, the small kernel Mc line and the large kernel V671 line. Using F_{2:3} families, a major QTL (*HKW9*) on chromosome 9 (bin 9.03-9.04) was detected, and accounted for more than

10% of phenotypic variation. Genetic analysis showed this QTL had a negative effect on HKW. In other words, even though the Mc line has smaller kernels, the Mc allele at the *HKW9* locus ($HKW9^{Mc}$) has a positive effect on kernel size, which exhibited transgressive segregation. To describe the construction of two NILs in more detail, we added the following information in the Methods. (*lines 597-605*)

*“Two elite maize inbred lines, Mc and V671, that develop small and large kernels, respectively, were used for mapping hundred kernel weight (HKW) QTL. Using $F_{2:3}$ families, a major QTL (*HKW9*) on chromosome 9 (bin 9.03-9.04) was detected, and accounted for more than 10% of phenotypic variation⁵. To construct near-isogenic lines (NILs), Mc was used as the recurrent parent, and the *HKW9* allele from V671 was introduced into Mc background through 4 generations of backcrossing followed by self-crossing. This resulted in two NILs, $HKW9^{Mc}$ and $HKW9^{V671}$, harboring the *HKW9* locus from Mc and V671, respectively.”*

In line 82, the sentence is a bit confusing to me. What do you mean by “were increased”? Does this mean that the difference between the two NILs was smaller in the past?

Response 2.2.2 We apologize for the confusion. To describe the result clearly, we have rephrased the description in the revised manuscript. (*lines 88-91*)

“Over successive years of field trials, the large-kernel NIL, $HKW9^{Mc}$, showed a significant increase in hundred kernel weight by 3.8%, in kernel length by 4.2%, in kernel width by 5.9% and in ear weight by 17.6%, compared to the small-kernel NIL, $HKW9^{V671}$.”

3) Line 105: Two candidate genes are strongly upregulated in the kernel. However, you only focused on *ZmEXPB15*. What about the second gene upregulated? It also encodes an expansin (*EXPB14*). Is there any change in its DNA sequence between the two NIL? Is it expressed in the nucellus too? Do you think it could be redundant with *ZmEXPB15*?

Response 2.3 We appreciate the reviewer’s careful comments. We agree, two out of 27 differentially expressed genes between the two NILs could be the candidates, that are highly and specifically expressed in nucellus of the developing kernels (please see the figure below). By resequencing the coding sequences, we found that *ZmEXPB14* contained only two nonsynonymous mutations in the non-conserved sites, whereas *ZmEXPB15* covered a 4-bp deletion in the conserved domain of the last coding exon (**newly added Supplementary Fig. 3**), therefore, our main focus was first on *ZmEXPB15*. Further, our association study supports the correlation of *ZmEXPB15* on the hundred kernel weight. Another important support was from the functional investigation of *ZmEXPB14* and *ZmEXPB15* by generating knock-out mutants through CRISPR/Cas9. Due to the high similarity, *ZmEXPB14* and *ZmEXPB15* were both edited in the identified three knockout lines (KO1, KO2, KO3), which showed significant decrease in HKW (**Fig. 2g**). To clarify the function of each expansin gene

which are tightly linked across the centromere, we tried hard to separate the two mutations in a large (>30,000) segregating population. Single *zmexpb15* mutant showed a significant decrease in HKW compared to the wild type, whereas *zmexpb14* mutation did not show an obvious effect on HKW (**newly added Fig. 1m and Supplementary Fig. 5c.** please also see the figure below). Importantly, *zmexpb14;zmexpb15* double mutant did not show an enhancement in the decrease of HKW compared to *zmexpb15* (**newly added Fig. 1m**), indicating that *ZmEXPB14* does not play a significant role in controlling kernel traits. Furthermore, the expression level of *ZmEXPB14* was not affected by the *zmexpb15* mutation (**newly added Supplementary Fig. 5d**), demonstrating that the decrease of HKW in *zmexpb15* was not related to *ZmEXPB14* expression. Therefore, we conclude that *ZmEXPB15* is the candidate of the QTL *HKW9*. We added the analysis of the candidate gene in the related context of the revised manuscript. (**lines 113-121, 139-150**)

“These two genes, Zm0001d045792 and Zm0001d045861, both encoding β -expansin, showed high (~98%) similarity in the coding regions and were designated as ZmEXPB14 and ZmEXPB15 hereafter, respectively. By resequencing the coding regions, we found that ZmEXPB14 contained only two nonsynonymous mutations in the non-conserved sites, whereas ZmEXPB15 covered a 4-bp deletion in the conserved domain of the last coding exon, leading to a frame shift of the last 60 amino acids and an addition of 24 residues in the C-terminus (Fig. 1h, Supplementary Fig. 3a, b). Thus, our main focus was first on ZmEXPB15.”

*“To further verify the candidate gene for the QTL *HKW9*, we generated knockout mutants for both *ZmEXPB14* and *ZmEXPB15* genes using a CRISPR/Cas9 strategy (Supplementary Fig. 5a). Only double mutants (KO1, KO2, KO3) were obtained with two genes being edited together due to their high similarity (Supplementary Fig. 5b). We next separated the *zmexpb14* and *zmexpb15* mutations by screening a large (>30,000) segregating population due to their close linkage across the centromere. Single *zmexpb14* mutant did not show an obvious change in HKW, whereas *zmexpb15* decreased the HKW significantly (Fig. 1m, Supplementary Fig. 5c). Importantly, the double *zmexpb14;zmexpb15* mutant did not show any enhancement in the HKW difference, compared to the *zmexpb15* single mutant in which *ZmEXPB14* expression was not altered (Fig. 1m, Supplementary Fig. 5d), indicating that *ZmEXPB15* is the major player for the HKW difference.”*

Figure legend. **a**, *ZmEXPB14* and *ZmEXPB15* are highly and specifically expressed in nucellus of developing kernel. The expression profile of *ZmEXPB14* and *ZmEXPB15* in three components of kernel is based on the public RNA-seq data¹². Nu0-Nu144: developing nucellus from 0 to 144 HAP (hours after pollination); Em12-Em36: developing embryos from 10 to 36 DAP; En6-En38: developing endosperms from 6 to 38 DAP. **b**, Quantitative analysis of the hundred kernel weight in *zmexpb14-3* single mutants showed no change in HKW. Number on each column is the sample size. **c**, The hundred kernel weight of *zmexpb15* single mutant was significantly decreased compared to wild type (WT), and the *zmexpb14;zmexpb15* double mutant showed a similar decrease. Number on each column is the sample size. **d**, The expression levels of *ZmEXPB14* was not altered in *zmexpb15-1* single mutant lines. The values in (**b-d**) are shown as the means \pm s.d., and the significance in (**b**) is estimated by one-way ANOVA, in (**d**) is estimated by a two-tailed Student's *t* test. ns, non-significance. The Tukey HSD test in (**c**) is used and statistical differences ($P < 0.05$) are indicated by different letters.

4) The authors paid a lot of attention on the difference in ZmEXP15 expression level. However, the ZmEXPB15 proteins are significantly different in the two NILs, due to a 4pb deletion and a shift in the ORF in HKW9V671. Where are the conserved domains of ZmEXPB15? Is the protein function potentially altered in HKW9V671?

Response 2.4 Thanks for your question. The 4-bp deletion in the last exon of *ZmEXPB15* caused a frame shift of the last 60 amino acids and an addition of 24 residues at the C-terminus (**Supplementary Fig. 3**). Based on alignment of *ZmEXPB15* sequences across different species, the last 60-AA region is conserved (please also see **Response 1.7**), therefore, we could not rule out the possibility that this 4-bp deletion might interrupt the function of *ZmEXPB15*. A direct proof for this

possibility would be a complementation test using this 4-bp deletion version of ZmEXPB15 protein in *zmexpb15* mutant. However, maize transformation is limited to few specific genetic backgrounds, and takes ~8 months, and another 8 months (2 generations) are needed to introgress the transgene into the *zmexpb15* mutant for the complementation. We regret that we are not able to provide this data in the limited time. Meanwhile, we showed that the SNPs in the promoter correlate with the expression level of ZmEXPB15 and also with the hundred kernel weight. Nevertheless, we have provided several lines of evidence showing that ZmEXPB15 is responsible for the difference of HKW between the two NILs, HKW9^{V671} and HKW9^{Mc}, and that ZmEXPB15 is an important gene controlling kernel weight and size.

5) Line 214: “GFP signal was enriched at the cell periphery, consistent with its predicted cell wall localization”. The protein localisation in the cell walls in Fig. 3D is not clear to me. Did the authors try to counter stain the cell walls or to perform a plasmolysis to assess the cell wall localization?

Response 2.5 Thank the reviewer for the constructive suggestion. We plasmolysed nucellus cells from *proZmEXPB15:ZmEXPB15-GFP* transgenic plants. Strong ZmEXPB15-GFP signals were found in cell wall, cytoplasm and nucleus (**updated Fig. 3d**, please also see the figure below). We thus rephrased the related description in the revised manuscript. (**lines 238-239**)

“After plasmolysis, the GFP signal was clearly detected in cell wall, cytoplasm and also nucleus (Fig. 3d).”

Figure legend. After plasmolysis, ZmEXPB15-GFP expression was observed in the cell wall (purple arrow), cytoplasm (yellow arrow) and nucleus (white arrow) of the nucellar cells in 4-DAP kernels. PI, propidium iodide staining. Merge, merge of GFP and PI images. Scale bar = 20 μ m.

6) Line 252: the authors studied the expression level of genes that are claimed to be PCD-related. Could the authors provide a reference showing that these genes are PCD-related?

Response 2.6 We agree, however since these potential PCD-genes have not been functionally characterized in maize, we refer to homologs in other plant species, and have added the references in the revised manuscript. (**lines 304-317**)

“Seed developmental PCD involves diverse classes of proteases, including cysteine proteases, serine proteases, and aspartic proteases, and special functions have been described for vacuolar proteases⁶. To search for the molecular basis for the effect of

ZmEXPB15 on nucellus PCD, we next compared the expression levels of some annotated protease genes between the two NILs, including vacuolar processing enzyme 4 (VPE4) homologous to barley VPE4 which promotes pericarp PCD^{7, 8}, cysteine protease 5 (CCP5) homologous to *Arabidopsis* xylem cysteine proteinase 1 (XCP1) and XCP2, which promote catabolism in tracheary elements during xylogenesis⁹, serine protease 1 (SER1) homologous to *Arabidopsis* senescence-associated gene SAG15, which function in protein degradation in senescing chloroplasts¹⁰, aspartyl protease 1 (AED1) and AED2 homologous to *Arabidopsis* aspartyl protease AED1, which may degrade apoplastic proteins¹¹. All these potentially PCD-related genes were expressed at significantly lower levels in 4-DAP kernels of HKW^{V671} than in HKW^{Mc} (Fig. 4d). This result is in line with the delayed PCD in HKW^{V671} in which *ZmEXPB15* was expressed at a lower level.”

7) Based on Figure 4, the difference between the two NILs in cell death and in endosperm expansion are very distinct in timing, suggesting they are not related.

Indeed, in Fig 4c, difference in TUNEL signal between the two NILs can only be seen at early stages and at 6DAP no difference is observed. However, at 6 DAP, no difference between the two NIL can be seen in the endosperm/nucellus ratio, the difference appears only at later stage (Fig 4b). This suggests that these two processes are not directly related because separated in time.

It could come from the fact that TUNEL is not the best proxy of cell death, as it only shows the presence of DNA fragmentation, which can happen very early in PCD. The use of other cell death markers such as Evans blue staining, FDA staining or cytological description on FFPE section or ultrastructural analyses by TEM would maybe give a better correlation between the difference in cell death and in endosperm growth and might be informative to understand more about the developmental origin of the phenotype described.

Response 2.7 We apologize for leaving out the statistical analysis for endosperm expansion in **previous Fig. 4b**. In fact, at 3 DAP and 6 DAP, the ratio of nucellus to endosperm area in the large-kernel line, HKW^{Mc}, was already significantly higher than that in the small-kernel line, HKW^{V671} (**updated Fig. 4i**, please also see the figure below). Therefore, the differences between the two NILs in cell death and in endosperm expansion are at a similar timing, suggesting they are not related. Further, we also investigated the appearance timing of the differences in nucellus PCD and endosperm expansion in *ZmEXPB15* KO and OE lines. In both cases, the TUNEL signals showed an obvious difference as early as 3 DAP, and the ratio of nucellus to endosperm area also showed difference as early as 3 DAP (**newly added Supplementary Fig. 8b-d**, please also see the figure in **Response 2.1**). These observations indicate that the timing of *ZmEXPB15* regulation of nucellus PCD is consistent with its role in early endosperm development.

In addition, following the reviewer’s suggestion, we performed Evan's blue staining to compare the PCD between the two NILs. As shown in the **newly added Supplementary Fig. 8e**, the large-kernel line, HKW^{Mc}, had deeper blue staining than the small-kernel line, HKW^{V671}, starting from 4 DAP. This observation is

consistent with the earlier nucellus PCD in HKW9^{Mc}. We also observed that *ZmEXPB15*-overexpression (OE1) kernels had deeper staining than non-transgenic ones as early as 2 DAP (**newly added Supplementary Fig. 8f**, please also see the figure in **Response 2.1**), indicating an enhanced nucellus cell death when *ZmEXPB15* expression was increased. This result suggests that the alteration in the nucellus/endosperm ratio between the two NILs could be directly related to the difference in the PCD timing of nucellus. We thus conclude that the large-kernel NIL initiates an earlier PCD and a faster cell death than the small-kernel NIL, leading to an early endosperm growth and the large-kernel phenotype. All these updates have been added in the related context of the revised manuscript. (*lines 269-303*)

Figure legend. Ratio of nucellus (orange box) to endosperm area (green box) in 0 to 12 DAP kernels of HKW9^{Mc} (M) and HKW9^{V671} (V). Longitudinal median sections with maximum area of the entire endosperm from more than 10 kernels for each line were examined and quantified. The values are shown as the mean \pm s.d., and the significance is estimated by a two-tailed Student's *t* test. ****P* < 0.001. ns, non-significance.

Minor comments:

Figure 7: indicate which *zmexpb15* mutant you used.

Response 2.8 The *zmexpb15-3* allele from KO3 was used. It's now been revised both in **updated Fig. 7** and the related text in the revised manuscript.

Line 95: change smaller by higher

Response 2.9 It's been changed as suggested.

Sup. Figure 1: change seeds by kernels.

Response 2.10 It's been changed as suggested.

Sup Figure 10: change c in the text by b.

Response 2.11 It's been changed as suggested.

Reviewer #3 (Remarks to the Author):

Sun et al., deciphered the functional implication of expansin gene (*ZmEXPB15*) in mediating nucellus degeneration, affecting endosperm growth and thus grain weight. In this study authors clearly demonstrated that *ZmNAC11* and *ZmNAC29* directly bind to *ZmEXPB15*. The knockout lines of these transcription factors (*ZmNAC11* and *ZmNAC29*), *ZmEXPB15*, and triple mutants exhibit delayed programmed cell death (PCD) of maternal tissue nucellar projection, wherein endosperm development is impacted with reduced kernel size and weight. The proposed work identified the regulation of the NAC-Expansin module but does not offer mechanistic insights. Also, the authors speculated that haplotype1 targeted for large-kernel NIL, HKWMc might be of use to increase grain weight in the breeding programs but do not offer direct evidence of breeding application in the manuscript.

Revealing further clarity will help to pinpoint the role of *ZmEXPB15* in mediating nucellus-endosperm growth dynamics and its impact on grain weight:

1. Although the Authors showed that *ZmNAC11*, *ZmNAC29*, and *ZmEXPB15* individual knock-out lines exhibit lowered hundred kernel weight, more so in the triple mutant exhibit, the exact molecular function of *ZmEXPB15* in mediating PCD of nucellus through potential cell wall loosening or cell wall degeneration not shown. Also, there were many proteases reported to be differentially regulated in these mutants. Bringing further clarity between the NAC-Expansin module and proteases will be helpful.

Response 3.1 Thank the reviewer for the constructive criticisms. We agree with the reviewer that the exact molecular function of *ZmEXPB15* in mediating PCD is still not clear. We originally detected lower expression of some PCD-related protease genes in the NIL line with lower *ZmEXPB15* level. Here we further observed that the *ZmEXPB15*-GFP signal accumulated not only in the cell wall, but also in nucleus of the nucellus from the *proZmEXPB15:ZmEXPB15*-GFP transgenic plants (**newly added Fig. 3d**, please also see the figure in **Response 2.5**). These results would help elucidate the function of *ZmEXPB15* in mediating nucellus PCD. While our discovery of differentially expressed proteases is presented as support of the cell death model, we feel that a molecular characterization of these proteases is beyond the scope of this manuscript. In addition, without more supporting data, we have removed the possibility that “*ZmEXPB15* may regulate the nucellus PCD through participation in potential cell wall loosening or cell wall degeneration” in the Discussion.

2. Authors claimed that *ZmEXPB15* contributes to the yield improvement of maize. To substantiate this claim yield data is a pre-requisite (plot yield) for CRISPR-Cas9 mutants and transgenic overexpression lines.

Response 3.2 We agree with the reviewer that plot yield data is more convincing to address the application value of *ZmEXPB15*. Our main focus of this manuscript is to characterize the function and its molecular mechanism of the QTL HKW9 candidate, *ZmEXPB15*, in controlling the early kernel development. The effect of *ZmEXPB15* on yield improvement requires repeated plot yield data. We thus toned down the related

claim by replacing “yield” with “kernel size” or “yield-component” in the related text of the revised manuscript.

3. Discussion is rather repetitive with results, offering a clear insight of molecular function of expansin with the onset of PCD will be helpful.

Response 3.3 Thank you for the constructive criticism. We tried to deepen the insight of the molecular function of *ZmEXPB15* with the onset of PCD, please refer to **Response 3.1** for the details. Following your suggestion, we have rephrased most parts of the Discussion, which are highlighted in blue in the revised manuscript (**lines 519-577**). Since the revised discussion is long, we did not paste it here.

Reviewer #4 (Remarks to the Author):

The manuscript “A NAC-EXPANSIN module enhances maize kernel size by controlling nucellus degeneration” by Qin Sun et al. describes the role of one EXPANSIN and two NAC genes in nucellus elimination during early maize kernel development. The authors identify the ZMEXPB15 gene as candidate for the HKW9 QTL, which affects kernel size and weight. Genetic and expression analyses show that this EXPANSIN is involved in the elimination of the nucellus and, as a consequence, affects the development of the endosperm and the overall kernel as well. Finally, the manuscript presents data suggesting that the transcription factors NAC11 and 29 work upstream ZMEXPB15 to facilitate the elimination of the nucellus.

The results are novel and interesting both in the of basic and applied research sphere. The manuscript is clear and well written.

Major points:

-My main concern regards the interpretation of the mutant phenotypes. The authors should check if ZMEXPB15 and NAC11 and 29 are involved in the expansion of the nucellus that follows fertilization, as the cell elimination phenotype might be only indirect. While some cells of the nucellus are eliminated, the others have to follow the expansion of the rest of the seed. Regardless of which hypothesis will turn out to be true, a more thorough phenotypic analysis has to be conducted to better image the processes of cell expansion and elimination.

Response 4.1 Thank the reviewer for the positive comments and constructive criticisms. Following the reviewer’s suggestion, we mainly checked the effect of *ZmEXPB15* on nucellus expansion, given that it works downstream of NAC11 and 29, and also could be more direct with cell expansion. By comparing the semi-thin sections of 3-DAP kernels of the two NILs, we found that the nucellus in the large-kernel line HKW9^{Mc} was thicker than that in HKW9^{V671} (**Newly added Fig. 4a-d**, please also refer to the figure below). Further, the cell area, cell length and width of the nucellar cells in HKW9^{Mc} were significantly larger than those in HKW9^{V671}, whereas the cell number was compatible (**Newly added Fig. 4e-g, Supplementary Fig. 8a**). These results indicate that *ZmEXPB15* could also promote

nucellus cell expansion. Together with our main finding that *ZmEXPB15* could promote nucellus elimination through mediating PCD and cell death, we speculate that *ZmEXPB15* regulates both cell expansion and elimination of the nucellus. We have added this new result in the related context of the revised manuscript.

“To investigate how ZmEXPB15 affects the nucellus development, we compared the nucellus size of the two NILs, and found that the nucellus in the large-kernel line HKW9^{Mc} with higher ZmEXPB15 expression was thicker than that in HKW9^{V671} (Fig. 4a). The cell area, cell length and width of nucellar cells in HKW9^{Mc} were significantly larger than those in HKW9^{V671}, while the cell number was compatible (Fig. 4b, c; Supplementary Fig. 8a). These observations indicated that ZmEXPB15 could promote nucellus cell expansion, leading to a larger nucellus volume.” (lines 259-265)

“Meanwhile, ZmEXPB15 also promotes nucellus cell expansion which is in line with its protein localization and property, leading to a larger nucellus volume. The dual role of ZmEXPB15 on nucellus cell expansion and elimination opens a new window to understand how the nucellus tissue growth coordinates with endosperm development.” (lines 522-526)

Figure legend. The large-kernel line HKW9^{Mc} has greater cell expansion. **a, b**, Longitudinal semi-thin section of 3-DAP kernels used for nucellus size measurement of HKW9^{Mc} and HKW9^{V671}. The black dotted lines indicate the region where nucellus thickness was measured. **c, d**, Enlarged image of the black boxed region in **a** and **b**, respectively. A row of nucellar cells are framed by black-dots in **c** and **d** to measure the cell area, cell length and cell width, and cell numbers. Scale bar = 200 µm in **(a-d)**. **e-i**, Quantification of nucellus thickness (**e**), cell number (**f**), cell areas (**g**), cell length (**h**) and cell width (**i**) of HKW9^{Mc} (M) and HKW9^{V671} (V). n = 6-7 biologically independent samples in **e, f**. The sample numbers in **g-i** refer to all cells counted along the black dotted lines from all independent samples. The values in **(e-i)** are shown as the means ± s.d., and the significance is estimated by a two-tailed Student’s *t* test. ***P* < 0.01, ****P* < 0.001, ns, non-significance.

-I believe that results from ZMEXPB15 over-expression lines would be meaningful only if the authors clearly show an effect on nucellus development

Response 4.2 Thank you for your consideration. Following your suggestion, we evaluated the nucellus development of *ZmEXPB15* overexpression (OE) line by investigating its PCD process. As shown in **newly added Supplementary Fig. 8d** (please also see **Response 2.1**), the OE line showed an earlier and stronger TUNEL signals in the nucellus at 3 and 6 DAP stages, compared to the control. Further, Evan's blue staining assay demonstrates an enhanced cell death in the *ZmEXPB15*-OE line as early as 2 DAP (**newly added Supplementary Fig. 8f**). As a consequence, the OE line had a lower ratio of nucellus to endosperm area at the stage of 3 DAP (**newly added Supplementary Fig. 8b**). These observations consistently demonstrate that enhanced *ZmEXPB15* promotes nucellus elimination, leading to increased endosperm expansion and consequently larger kernel size and weight.

-Fig 4 and Fig6: the results obtained by measuring nucellus and endosperm areas strongly depend on the section used. How do the authors pick their sections? How do they make sure to have longitudinal-mid sections?

Response 4.3 The longitudinal median sections were selected for measurement when they had the maximum area of the ovules at 0 DAP or the endosperm at 2-3 DAP. For the kernels at 4 DAP and later stages, the maximum area of embryos was used for the reference, since the entire endosperm was too large to judge accurately. More than 10 kernels at each stage were sectioned and measured for quantification.

-Fig 4: The authors should have performed the same analyses also with *Zmexp15* CRISPR lines.

Response 4.4 Thank you for your valuable suggestion. We analyzed the nucellus elimination of *zmexpb15* KO line shown in **newly added Supplementary Fig. 8b, c** (please also see **Response 2.1**). Briefly, the TUNEL signals in the KO nucellus was nearly absent at 3 DAP, whereas the wild-type nucellus already showed strong TUNEL signals at this stage, and the difference was also visible at the stage of 6 DAP. Further, the KO line had a higher ratio of nucellus to endosperm area than the control as early as the stage of 3 DAP. Thus, *zmexpb15* KO line also delayed in nucellus elimination, resulting in a delay in endosperm development and consequently the decreased kernel size and weight.

-Data on the direct binding of NAC 11 and 29 on the ZMEXPB15 promoter are not conclusive. A Chip experiment is necessary to claim direct binding. I understand that it is not an easy experiment, therefore I would ask the authors to adjust model and text accordingly. Furthermore, I do not see signs of competition in some of the EMSA analyses.

Response 4.5 We highly appreciate the reviewer's consideration. We agree with the reviewer that in vivo ChIP data is necessary to claim a direct binding, and thus toned down the model and the text accordingly by changing "directly bind" to "could bind" in the revised manuscript. We also repeated the competitive EMSA experiment twice

and got a consistent result. As shown in the **updated Fig. 5d** (please also see the figure below), both the *ZmNAC11* and *ZmNAC29* recombinant proteins show obviously stronger binding to the *ZmEXPB15* promoter fragment from the large-kernel line HKW9^{Mc} than that the small-kernel line HKW9^{V671}.

Figure legend. DNA binding affinities of the recombinant *ZmNAC11* and *ZmNAC29* proteins on the CACG motif-containing promoter regions of HKW9^{Mc} and HKW9^{V671} detected by electrophoresis mobility shift assays (EMSAs). *ZmNAC11* and *ZmNAC29* bind more strongly to the promoter fragments from the large kernel line HKW9^{Mc} than to those of the small kernel line HKW9^{V671}. The unlabeled intact probes were used for competition. The experiment was repeated two times with a similar result.

-The expression pattern of NAC 11 and 29 should be more carefully characterized by RNA in situ hybridization experiments.

Response 4.6 Following the reviewer’s suggestion, we performed RNA in situ hybridization against *ZmNAC11* and *ZmNAC29*, and obtained a nucellus-enriched expression pattern (**newly added Fig. 5c**), please see the figure below.

Figure legend. In situ localization of *ZmNAC11* and *ZmNAC29* transcripts in the developing kernels at 4 and 6 DAP. Positive signals were detected in the nucellus (Nu) and endosperm (En) using the *ZmNAC11* and *ZmNAC29* antisense probes. The controls were performed using *ZmNAC11* and *ZmNAC29* sense probes. Scale bars = 50 μ m.

Minor Points:

-The process of nucellus death is now referred to as “cell elimination” because it leads to the full degeneration of the cell including the cell wall.

Response 4.7 We changed the “nucellus degradation” to “nucellus elimination” in full text of the revised manuscript.

-Line 112: The data presented do not demonstrate that the differential expression is due to the promoter SNPs.

Response 4.8 Thanks for your reminder. We have rephrased this sentence. (*lines 123-125*)

“Consistently, the *ZmEXPB15* transcript was found to be differentially expressed between the two NILs from an early stage (~2 DAP) of kernel development, with a higher *ZmEXPB15* expression in the large-kernel NIL, *HKW9^{Mc}*”

References cited in the response letter

1. Radchuk, V. et al. *Jekyll* encodes a novel protein involved in the sexual reproduction of barley. *Plant Cell* **18**, 1652-1666 (2006).
2. Xu, W. et al. Endosperm and nucellus develop antagonistically in *Arabidopsis* seeds. *Plant Cell* **28**, 1343-1360 (2016).
3. Wu, H. M., Cheung, A. Y. Programmed cell death in plant reproduction. *Plant Mol. Biol.* **44**, 267-281 (2000).
4. Dominguez, F., Cejudo, F. J. Programmed cell death (PCD): an essential process of cereal seed development and germination. *Front. Plant Sci.* **5**, 366 (2014).
5. Liu, Y. et al. Genetic analysis and major QTL detection for maize kernel size and weight in multi-environments. *Theor. Appl. Genet.* **127**, 1019-1037 (2014).
6. Lopez-Fernandez, M. P., Maldonado, S. Programmed cell death in seeds of angiosperms. *J. Integr. Plant Biol.* **57**, 996-1002 (2015).
7. Radchuk, V. et al. Vacuolar processing enzyme 4 contributes to maternal control of grain size in barley by executing programmed cell death in the pericarp. *New Phytol.* **218**, 1127-1142 (2018).
8. Radchuk, V. et al. Grain filling in barley relies on developmentally controlled programmed cell death. *Commun Biol.* **4**, 428 (2021).
9. Avci, U., Petzold, H., E, Ismail, I. O., Beers, E. P., Haigler, C. H. Cysteine proteases XCP1 and XCP2 aid micro-autolysis within the intact central vacuole during xylogenesis in *Arabidopsis* roots. *Plant J.* **56**, 303-315 (2008).
10. Nakabayashi, K. et al. Identification of *clp* genes expressed in senescing *Arabidopsis* leaves. *Plant Cell Physiol.* **40**, 504-514 (1999).
11. Till, C. J. et al. The *Arabidopsis thaliana* N-recogin E3 ligase PROTEOLYSIS1 influences the immune response. *Plant Direct.* **3**, e00194 (2019).
12. Yi, F. et al. High temporal-resolution transcriptome landscape of early maize seed development. *Plant Cell* **31**, 974-992 (2019).

Reviewers' Comments:

Reviewer #1:

Remarks to the Author:

The authors have done a superb job addressing all the reviewers' comments and I am satisfied with this revision. The work is of high quality and would be a pleasure to read.

Reviewer #2:

Remarks to the Author:

The manuscript "A NAC-EXPANSIN module enhances maize kernel size by controlling nucellus elimination" has been greatly improved compared to the previous version. The authors answered clearly the questions raised and provided a large amount of new results that significantly enhance the quality of the manuscript. Several key results have been added including i) a description of ZmEXPB15 KO and OE phenotypes in the nucellus ii) a description of ZmEXPB14, the second expansin candidate in the QTL and the rationale behind the choice of ZmEXPB15, iii) a description of the difference in nucellus expansion between the two NILs iv) the confirmation by Evans Blue staining of a difference between the two NILs in nucellus cell death v) the proof of a direct binding of ZmNAC11 and ZmNAC29 on ZmEXPB15 promoter vi) in situ hybridization against ZmNAC29 and ZmNAC11 showing their expression in both nucellus and endosperm.

Although I am overall very satisfied by the changes made, they raise an additional question:

1) You provide strong evidences indicating that ZmNAC11 and ZmNAC29 regulate ZmEXPB15 expression. In addition, *zmnac11* and *zmnac29* display similar HKW as *zmexpb15* (Fig. 7b) and similar effect on nucellus elimination (Fig 6). Altogether, these results suggest that *zmnac11* and *ZmNAC29* regulate HKW through ZmEXPB15. However, the triple mutant *zmnac11 zmnac29 zmexpb15* shows an additive effect on HKW. As discussed by the authors, this rather suggests that additional targets of ZmNAC11 and ZmNAC29 are involved in HKW. In situ hybridization against ZmNAC11 and *zmnac29* show that both genes are not only expressed in the nucellus but also in the endosperm. Which part of HKW reduction in *zmnac11*, *zmnac29* and *zmnac11 zmnac29 zmexpb15* comes really from defects in the nucellus?

A straightforward experiment to assess this would be to cross these mutants with wild type pollen (mutant X WT). If the reduction in HKW comes from the nucellus, it should be unchanged when crossed with WT pollen. At least a part of the phenotype should still be visible

Did the authors do these crosses? I am aware that if it is not the case, this additional experiment could take 4-5 months. I do not know how compatible this would be with the publication plan.

I have also a concern about the discussion:

2) In the discussion you say, "ZmEXPB15 encodes an expansin protein, with an expected cell wall localization, whereas PCD happens in the nucleus and cytoplasm, raising a concern how this is coordinated. We found that the localization of a ZmEXPB15 GFP fusion protein was not only in the cell wall, but also in cytoplasm and nucleus of nucellus cells (Fig. 3d),

helping to elucidate the molecular function of ZmEXPB15 in mediating PCD process". Here, I would attenuate. I do not think EXPB15 localization in the nucleus and cytoplasm tell something about its molecular function in mediating PCD process. If EXPB15 would be directly involved in PCD, we would expect cell death symptoms in the ubiquitous OE line, which seems not to be the case, advocating for a more indirect effect on PCD. In addition, expansin are known to act in the cell walls (doi: 10.1007/s00299-016-1948-4) and we cannot exclude that this localization is either artifactual or non-relevant biologically.

Minor point: In lines 464-470 ZmEXPB15 is written ZmEXP15

Reviewer #4:

Remarks to the Author:

The revised version of the manuscript "A NAC-EXPANSIN module enhances maize kernel size by controlling nucellus degeneration" by Qin Sun et al. answers all the points I have raised in my first review and I believe it should be accepted for publication.

Reviewer #1 (Remarks to the Author):

The authors have done a superb job addressing all the reviewers' comments and I am satisfied with this revision. The work is of high quality and would be a pleasure to read.

Response1. We greatly appreciate the reviewer for the positive comments.

[Editor: **Reviewer #3** is unavailable. Reviewer #1 helps to comment.]

Response3. Thank Reviewer #1 for his great help to comment.

Reviewer #4 (Remarks to the Author):

The revised version of the manuscript “A NAC-EXPANSIN module enhances maize kernel size by controlling nucellus degeneration” by Qin Sun et al. answers all the points I have raised in my first review and I believe it should be accepted for publication.

Response4. We appreciate the reviewer for the positive comments.

Reviewer #2 (Remarks to the Author):

The manuscript ‘A NAC-EXPANSIN’ module enhances maize kernel size by controlling nucellus elimination” has been greatly improved compared to the previous version. The authors answered clearly the questions raised and provided a large amount of new results that significantly enhance the quality of the manuscript. Several key results have been added including i) a description of ZmEXPB15 KO and OE phenotypes in the nucellus ii) a description of ZmEXPB14, the second expansin candidate in the QTL and the rationale behind the choice of ZmEXPB15, iii) a description of the difference in nucellus expansion between the two NILs iv) the confirmation by Evans Blue staining of a difference between the two NILs in nucellus cell death v) the proof of a direct binding of ZmNAC11 and ZmNAC29 on ZmEXPB15 promoter vi) in situ hybridization against ZmNAC29 and ZmNAC11 showing their expression in both nucellus and endosperm.

Although I am overall very satisfied by the changes made, they raise an additional question:

1) You provide strong evidences indicating that *ZmNAC11* and *zmNAC29* regulate *ZmEXPB15* expression. In addition, *zmnac11* and *zmnac29* display similar HKW as *zmexpb15* (Fig. 7b) and similar effect on nucellus elimination (Fig 6). Altogether, these results suggest that *zmNAC11* and *ZmNAC29* regulate HKW through *ZmEXPB15*

However, the triple mutant *zmnac11 zmnac29 zmexpb15* shows an additive effect on HKW. As discussed by the authors, this rather suggest that additional targets of *ZmNAC11* and *ZmNAC29* are involved in HKW. In situ hybridization against *ZmNAC11* and *zmNAC29* show that both genes are not only expressed in the nucellus but also in the endosperm.

Which part of HKW reduction in *zmnac11*, *zmnac29* and *zmnac11 zmnac29 zmexpb15* comes really from defects in the nucellus?

A straightforward experiment to assess this would be to cross these mutants with wild type pollen (mutant X WT). If the reduction in HKW comes from the nucellus, it should be unchanged when crossed with WT pollen. At least a part of the phenotype should still be visible

Did the authors do these crosses? I am aware that if it is not the case, this additional experiment could take 4-5 months. I do not know how compatible this would be with the publication plan.

Response 2.1 We appreciate the reviewer for the constructive suggestion and consideration. Following the reviewer's comments, we investigated the maternal effect of the *zmnac11* and *zmnac29* mutations. As shown in the **newly added Supplementary Fig. 13** (please also see the figure **a, b** below), the kernel size and weight of F₁ kernels from WT ear pollinated with the *zmnac11-1; zmnac29-1* (*nac*) double mutant were significantly larger than those of *nac* ear pollinated with WT pollen. Therefore, *ZmNAC11* and *ZmNAC29* also regulate kernel size and weight in a maternal manner, similar to *ZmEXPB15*. Further, we also investigated whether the HKW reduction in *zmnac11; zmnac29; zmexpb15* (triple) mutant comes really from the

nucellus defects. As shown in the figure below (**c**, **d**), the kernel size and hundred kernel weight (HKW) of the heterozygous kernels from the triple mutant pollinated by WT pollen were similar to those from the selfed homozygous triple mutant, demonstrating that the reduction of the HKW in the triple mutant was mainly caused by the nucellus defects. Thus, the results are compatible with the conclusions in our manuscript. The maternal effects of the two NAC genes on kernel development was updated in the revised manuscript, *lines 428-430*,

“In addition, the hundred kernel weight of F₁ kernels from WT ear pollinated with the nac double mutant was significantly larger than that of nac ear pollinated with WT pollen (Supplementary Fig. 13a, b). The reciprocal cross results demonstrate that ZmNAC11 and ZmNAC29 control kernel development also maternally.”

Figure legend. **a**, The F₁ kernels of wild-type (WT) pollinated by *zmnac11-1; zmnac29-1* double mutant (*nac*) was obviously larger than those of *zmnac11-1; zmnac29-1* pollinated by WT. **b**, Quantitative analysis of the hundred kernel weight of F₁ kernels from the WT and *nac* reciprocal crosses. **c**, The *zmnac11-2; zmnac29-1; zmexpb15-3* (*triple*) mutant selfed homozygous kernels and outcrossed heterozygous kernels with WT pollen have similar phenotypes. **d**, Quantitative analysis of the hundred kernel weight of the *zmnac11-2; zmnac29-1; zmexpb15-3* (*triple*) mutant selfed homozygous kernels and outcrossed heterozygous kernels with WT pollen. The values in (**b**, **d**)

are shown as the means \pm s.d. (standard deviation), and the significance in **(b, d)** is estimated by one-way ANOVA. *** $P < 0.001$, ns, non-significance. Numbers on the bottom of each column are the sample size. Scale bar = 1 cm in **(a, c)**.

I have also a concern about the discussion:

2) In the discussion you say, “ZmEXPB15 encodes an expansin protein, with an expected cell wall localization, whereas PCD happens in the nucleus and cytoplasm, raising a concern how this is coordinated. We found that the localization of a ZmEXPB15 GFP fusion protein was not only in the cell wall, but also in cytoplasm and nucleus of nucellus cells (Fig. 3d), helping to elucidate the molecular function of ZmEXPB15 in mediating PCD process”.

Here, I would attenuate. I do not think EXPB15 localization in the nucleus and cytoplasm tell something about its molecular function in mediating PCD process. If EXPB15 would be directly involved in PCD, we would expect cell death symptoms in the ubiquitous OE line, which seems not to be the case, advocating for a more indirect effect on PCD. In addition, expansin are known to act in the cell walls (doi: 10.1007/s00299-016-1948-4) and we cannot exclude that this localization is either artifactual or non-relevant biologically.

Response 2.2 Thank the reviewer for the constructive suggestion. We fully agree that the nucleus and cytoplasm localization of ZmEXPB15 protein could not tell much about its molecular function in mediating PCD process, and thus attenuate our speculation. We revised the related part in Discussion of the revised manuscript, *lines 548-555*,

“ZmEXPB15 encodes an expansin protein, which participates in various biological processes by affecting the loosening of the cell wall^{36, 37}. The ZmEXPB15 protein localized on cell wall and explains its role in nucellus cell expansion (Fig. 3d, Fig. 4a-c). Our main finding on its role in nucellus PCD process opens a new window on expansin protein function, which is supported by the cytoplasm and nucleus localization of the GFP-ZmEXPB15 fusion protein (Fig. 3d). Further studies are required to elucidate the underlying mechanism on how ZmEXPB15 triggers the initiation of PCD process.”

In addition, we performed transient expression of ZmEXPB15-GFP fusion protein, and also found its nucleus and cytoplasm localization in onion epidermal cells and maize protoplasts (please see the figure below). These results support that ZmEXPB15 protein very likely not only localize in cell wall, but also in nucleus and cytoplasm. This would be our start point to elucidate the mechanism of its role in PCD process in future studies.

Figure legend. Subcellular localization of ZmEXPB15-GFP signal in onion epidermal cells (a) and maize leaf protoplasts (b). ZmEXPB15-GFP signal was localized in the nucleus (white arrow), cytoplasm (orange arrow) and cell wall (magenta arrow) of onion epidermal cells after plasmolysis. Maize protoplasts also show ZmEXPB15-GFP localization in the nucleus and cytoplasm. PI: Propidium iodide; BF: Bright field; Merge: Merges GFP and propidium iodide images (a); Merge: Merges GFP and bright field images (b). Scale bar = 50 μm in (a); 20 μm in (b).

Minor point: In lines 464-470 ZmEXPB15 is written ZmEXP15

Response Thanks for your reminder. It's been changed as suggested.

References cited in the response letter

36. Cosgrove, D. J. Plant expansins: diversity and interactions with plant cell walls. *Curr. Opin. Plant Biol.* **25**, 162-172 (2015).
37. Marowa P, Ding A, Kong Y. Expansins: roles in plant growth and potential applications in crop improvement. *Plant Cell Rep.* **35**, 949-965 (2016).

Reviewers' Comments:

Reviewer #2:

Remarks to the Author:

I am now fully satisfied by the revised version of the article "A NAC-EXPANSIN module enhances maize kernel size by controlling nucellus elimination". The authors positively addressed all the comments I had.